# Acetylation-dependent coupling between G6PD activity and apoptotic signaling

Fang Wu[1], Natali H. Muskat[1], Inbar Dvilansky[2], Omri Koren[3], Anat Shahar[4], Roi Gazit [3,5], Natalie Elia[2,5] & Eyal Arbely [1,2,5] ✉

Lysine acetylation has been discovered in thousands of non-histone human proteins, including most metabolic enzymes. Deciphering the functions of acetylation is key to understanding how metabolic cues mediate metabolic enzyme regulation and cellular signaling. Glucose-6-phosphate dehydrogenase (G6PD), the rate-limiting enzyme in the pentose phosphate pathway, is acetylated on multiple lysine residues. Using site-specifically acetylated G6PD, we show that acetylation can activate (AcK89) and inhibit (AcK403) G6PD. Acetylation-dependent inactivation is explained by structural studies showing distortion of the dimeric structure and active site of G6PD. We provide evidence for acetylation-dependent K95/97 ubiquitylation of G6PD and Y503 phosphorylation, as well as interaction with p53 and induction of early apoptotic events. Notably, we found that the acetylation of a single lysine residue coordinates diverse acetylation-dependent processes. Our data provide an example of the complex roles of acetylation as a posttranslational modification that orchestrates the regulation of enzymatic activity, posttranslational modifications, and apoptotic signaling.

Lysine acetylation, the post-translational addition of an acetyl group to the ε-amine of lysine residues, is a ubiquitous posttranslational modification (PTM) found on thousands of non-histone proteins. Originally thought to be involved in transcription regulation via the acetylation of histone proteins, acetylation is now additionally associated with metabolism, cellular signaling, and other major cellular functions. Enzymatic acetylation is dynamically regulated by lysine acetyltransferases (KATs) and lysine deacetylases (KDACs), which catalyze the attachment and hydrolysis of the acetyl group, respectively. By altering the charge and size of the modified lysine, acetylation may affect protein stability, subcellular localization, and interaction with other macromolecules in the cell[1-4]. Of particular importance to this study, acetylation of metabolic enzymes has been identified as an evolutionarily conserved PTM[5,6], and was found to regulate key enzymes in central metabolic pathways in eukaryotic cells[1,2,7,8].

However, the functional roles of most lysine acetylation sites in metabolic enzymes and their effect on non-metabolic processes are mostly unknown.

The pentose phosphate pathway (PPP) plays a significant role in the biosynthesis of nucleotides and is a major pathway for producing cellular reducing power. Glucose-6-phosphate dehydrogenase (G6PD) catalyzes the first and rate-limiting reaction in the PPP; the oxidation of glucose-6-phosphate (G6P) to 6-phosphogluconolactone with concomitant reduction of nicotinamide adenine dinucleotide phosphate (NADP$^+$) to NADPH. G6PD is a major source of cellular NADPH and has a primary role in controlling the metabolic flux through the PPP. Thus, abnormal G6PD activity is associated with numerous pathophysiological cellular alterations and diseases, including clinical G6PD deficiency (the most common blood disorder, with a prevalence of 1 in 20), infection, and cancer[9-11].

[1]Department of Chemistry, Ben-Gurion University of the Negev, Beer-Sheva 8410501, Israel. [2]Department of Life Sciences, Ben-Gurion University of the Negev, Beer-Sheva 8410501, Israel. [3]The Shraga Segal Department of Microbiology, Immunology and Genetics, Ben-Gurion University of the Negev, Beer-Sheva 8410501, Israel. [4]Macromolecular Crystallography Research Center (MCRC), Ilse Katz Institute for Nanoscale Science & Technology, Ben-Gurion University of the Negev, Beer-Sheva 8410501, Israel. [5]The National Institute for Biotechnology in the Negev, Ben-Gurion University of the Negev, Beer-Sheva 8410501, Israel. ✉e-mail: arbely@bgu.ac.il

G6PD is catalytically active as a homodimer or homotetramer. Each monomer has one G6P binding site (i.e., the catalytic site) and can bind two NADP$^+$ molecules; one is close to the G6P binding site, and the other is the structural NADP$^+$ binding site. The binding of structural NADP$^+$ stabilizes G6PD and promotes the formation of catalytically active homomers[12,13]. G6PD activity can also be regulated by PTMs such as O-GlcNAcylation and phosphorylation[14–17]. Importantly, proteomic analyses of different species have found that G6PD is post-translationally modified by lysine acetylation at multiple sites[18–33]. Specifically, acetylation of G6PD at position K403 (within the structural NADP$^+$ binding site) or K171 (within the active site) inhibits G6PD activity. At the same time, Sirt2 can deacetylate and reactivate mono-meric K403-acetylated (AcK403) G6PD to promote NADPH production in leukemia cells[33–37].

Our current understanding of cellular events downstream of acetylation is mainly derived from Lys-to-Gln mutational analyses[34,35,38]. However, glutamine is electrostatically and structurally different from acetylated lysine (AcK) and may not faithfully mimic the functional and structural effects of lysine acetylation. Consequently, the regulatory roles of individual lysine acetylation events are still elusive. These limitations can be removed by using genetic code expansion (GCE) technology to genetically encode the incorporation of non-canonical amino acids (e.g., AcK) into proteins expressed in live biological system[39–42]. To express site-specifically acetylated proteins, an evolved aminoacyl-tRNA synthetase is employed, to charge its cognate amber suppressor tRNA (tRNA$_{CUA}$) with AcK. The charged tRNA$_{CUA}$ enables the cotranslational incorporation of AcK in response to an in-frame UAG (amber) stop codon mutation placed at essentially any position along the expressed protein of interest, most commonly, at lysine residue already known to be acetylated from previous data.

To obtain an unbiased and complete insight into the molecular mechanism of G6PD regulation by acetylation, we used GCE to study in vitro and in cultured mammalian cells full-length human G6PD, site-specifically acetylated at each of the confirmed acetylation sites (namely: K89, K95, K97, K171, K386, K403, K408, K432, and K497). We show that acetylation can not only inhibit G6PD (K403 and K171 acetylation) but can also increase catalytic activity following K89 acetylation. Crystal structures of AcK89 and AcK403 G6PD suggest a mechanism for K89 acetylation-dependent ubiquitylation of G6PD and provide a detailed molecular mechanism for the inactivation of AcK403 G6PD. In addition, we found that K403 acetylation modulates Fyn-dependent phosphorylation of G6PD and interaction with p53. Importantly, K403 acetylation promotes the induction of apoptotic signaling, providing a unique example of crosstalk between metabolism and apoptotic evens that is mediated by the acetylation of a metabolic enzyme.

## Results

### G6PD activity is dynamically regulated by acetylation in cells

To determine the general (i.e., not site-specific) effect of acetylation on endogenous G6PD activity, we treated HEK293T and HCT116 cells with the lysine deacetylase inhibitors (KDACi) nicotinamide [NAM, an inhibitor of nicotinamide adenine dinucleotide (NAD$^+$)-dependent sirtuins] and suberoylanilide hydroxamic acid (SAHA, an inhibitor of Zn$^{2+}$-dependent deacetylases). G6PD activity in cell lysates decreased by ~50% following treatment with SAHA, but treatment with NAM or NAM and SAHA was significantly more potent relative to SAHA alone (Fig. 1a). These data suggest that G6PD activity is modulated by lysine acetylation and that NAD$^+$-dependent sirtuins play a major role in regulating G6PD acetylation level.

To study the functional roles of individual acetylation sites, we identified lysine residues that were found to be acetylated in human cells by at least two independent proteomic studies[18–23]. These acetylation sites were evaluated based on their position in the three-dimensional structure of human G6PD (Fig. 1b and Supplementary

Fig. 1a), their evolutionary and structural conservation (Supplementary Fig. 1b), whether a given position was found to be acetylated in other species[24–32], and bioinformatic prediction of the pathogenicity of a mutation of each lysine residue (Supplementary Fig. 1c, d). Based on these analyses, we selected nine putative acetylation sites: K89, K95, K97, K171, K386, K403, K408, K432, K497 (Fig. 1b).

The nine C-terminally Flag-tagged and monoacetylated G6PD variants were expressed in G6PD knocked-down HEK293T and HCT116 cells by genetically encoding the site-specific incorporation of AcK (Supplementary Fig. 1e)[39,43,44]. To compare between acetylated and non-acetylated G6PD, we also included, as control, two pseudo-wild type (pWT) versions of G6PD in which AcK was incorporated into a permissive site (Q83 or N414). In pWT G6PD, Q83, or N414 are mutated to AcK. Therefore, these variants are expressed at the same level as other physiologically relevant acetylated variants and can serve as a non-acetylated control. In addition, the use of pWT as a control takes into account any unknown effects of amber suppression or expression of truncated proteins on cell physiology. Immunoblot quantification of total cell lysates using an antibody against the C′-Flag tag confirmed the expression of full-length acetylated G6PD variants and two pWT versions, with minor variability in expression levels (Supplementary Fig. 1e). High expression of C′-truncated G6PD was detected only for AcK497. In addition, all proteins were predominantly cytosolic, as previously reported for wild-type (WT) G6PD (Supplementary Fig. 2a, b)[45].

To evaluate the acetylation state of exogenously expressed acetylated G6PD, all G6PD variants were expressed in HEK293T cells cultured in the presence or absence of KDACi. Immunoblot analysis using anti-acetylated lysine (anti-AcK) antibodies recognized all the acetylated G6PD variants but not WT G6PD, further confirming the UAG suppression-dependent expression of acetylated G6PD variants (Fig. 1c). Moreover, LC-MS/MS analyses of immunopurified G6PD expressed in the presence or absence of KDACi confirmed that the genetically encoded acetylated lysine residues are the most abundant acetylation sites, with no additional acetylation-dependent acetylation events (Supplementary Table 1). All acetylation sites were recognized by two commercially available anti-AcK antibodies in a context-dependent manner (Supplementary Fig. 2c)[3,46]. In general, we found an increase in acetylation level following KDAC inhibition, mainly at positions K89, K95, K386, K403, K408, and K432 (Fig. 1c). Thus, these acetylation sites are dynamically regulated by endogenous KDACs, suggesting a possible physiological relevance for the acetylation at these positions.

### Acetylation affects G6PD activity and stability

The effect of site-specific acetylation on G6PD enzymatic activity was initially studied in cleared cell lysates using a continuous enzyme-coupled assay[47]. Knockdown of endogenous G6PD and efficient expression of sh-resistant G6PD TAG mutants ensured that the measured activity reflects the activity of the exogenous acetylated form (Supplementary Fig. 2d, e). The catalytic activity of AcK497 G6PD was not determined in cell lysates, since its truncated form (G6PD$^{(1-496)}$) displayed residual catalytic activity (Supplementary Fig. 2f).

Catalytic activity measurements under $V_{max}$ conditions were performed in lysates of HEK293T and HCT116 cells cultured in the presence or absence of KDACi (Fig. 1d). The catalytic activity of the two pWT versions was comparable and independent of KDAC inhibition. We also found that except for AcK171 G6PD—and to some extent, AcK408—the catalytic activity of acetylated G6PD variants expressed in cells cultured without KDACi was similar to the activity of the two pWT versions, suggesting either no effect of acetylation on activity or efficient deacetylation by endogenous KDACs. The KDACi-independent inhibitory effect of K171 acetylation can be explained by the position of catalytic K171 within the active site, which may protect it from enzymatic deacetylation. In the presence of KDACi,

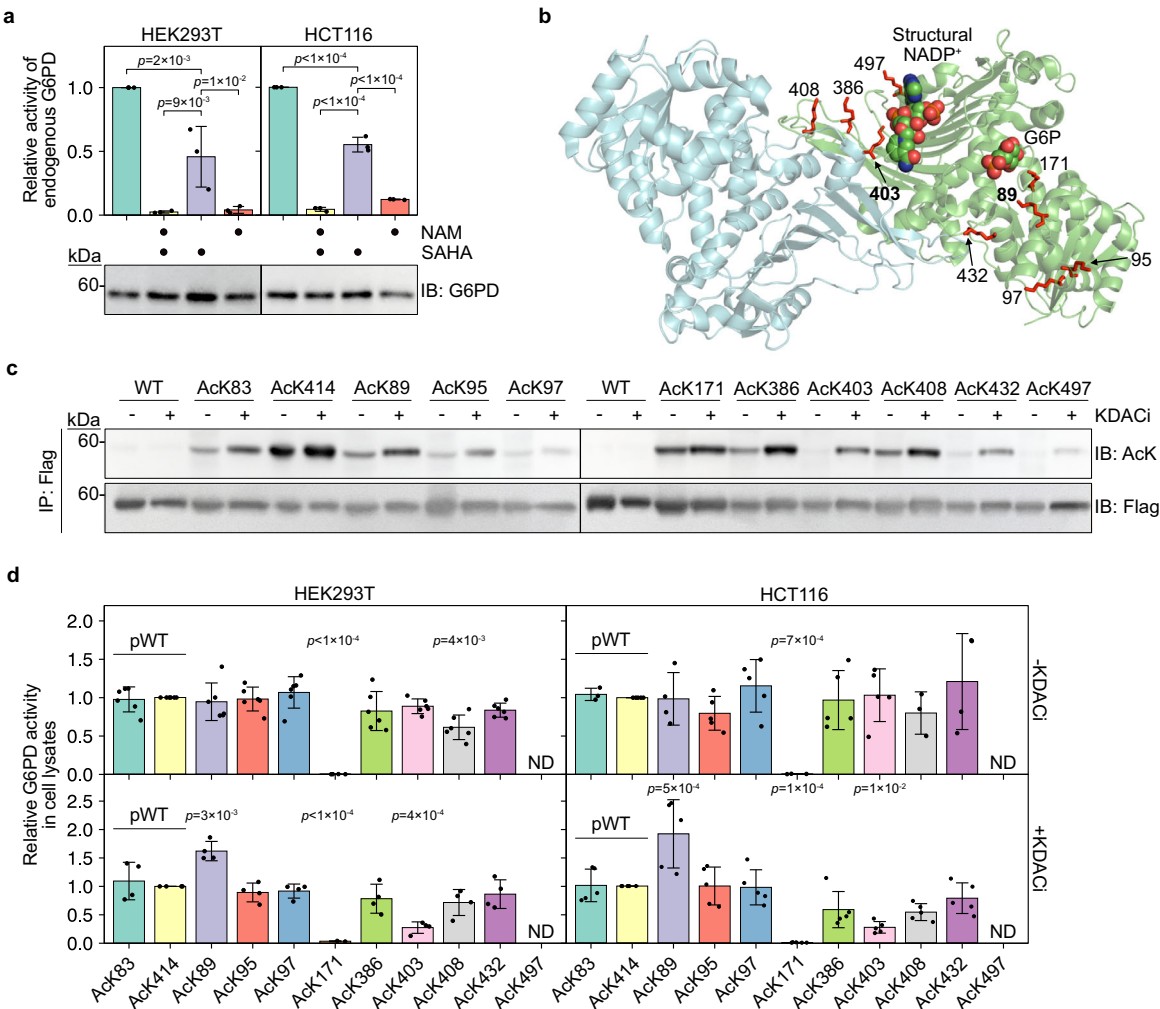

**Fig. 1 | Site-specific acetylation modulates the catalytic activity of G6PD in cultured mammalian cells. a** Inhibition of KDACs downregulates G6PD activity. Bars represent the relative $V_{max}$ of endogenous G6PD measured in cleared lysates of HEK293T and HCT116 cells cultured in the presence or absence of KDACi. Data were analyzed using one-way ANOVA followed by Tukey's post hoc test and are presented as mean values ± SD; $n = 3$ biologically independent samples. **b** Crystal structure of dimeric human G6PD (PDB ID: 6E08). Putative lysine acetylation sites selected for this study are presented in sticks model. G6P (predicted position) and structural $NADP^+$ are displayed as spheres. The main focus of this study is on residues K89 and K403, which are highlighted in bold. **c** Acetylation of specific lysine residues is dynamic and differentially regulated by KDACs. Immunoblots show the acetylation level of immunopurified acetylated G6PD-Flag variants, expressed in HEK293T cells cultured in the presence (+) or absence (−) of KDACi. **d** Site-specific acetylation modulates G6PD activity. Bars represent the relative $V_{max}$ of exogenous full-length G6PD measured in cell lysate. Indicated acetylated G6PD variants were expressed in HEK293T (left) and HCT116 (right) cells cultured in the absence (top) or presence (bottom) of KDACi. G6PD activity was normalized to immunoblot intensities of expressed G6PD and displayed relative to AcK414 (pWT G6PD). Data were analyzed using one-way ANOVA followed by Tukey's post hoc test and are presented as mean values ± SD; $n = 4$ or 6 (HEK293T), 3, 4, or 5 (HCT116) biologically independent samples. ND, not determined. Source data are provided as a Source Data file.

acetylation of K403 significantly reduced G6PD activity, and acetylation of K89 increased the catalytic activity of G6PD under $V_{max}$ conditions (Fig. 1d). In agreement with our measurements and previous studies, the K403Q mutation abolished enzymatic activity, while the activity of K403R G6PD was similar to WT G6PD (Supplementary Fig. 2g)[34]. However, the K89Q mutation did not affect G6PD activity, indicating that glutamine is not a good mimic for acetylation at position 89. Collectively, these observations suggest that acetylation/ deacetylation balance at K89, K171, K386, K403, K408, and K432 modulates G6PD enzymatic activity. Acetylation of catalytic K171 is expected to render the enzyme inactive[34], and K95/K97 did not affect enzymatic activity. Therefore, we decided not to further study these three acetylation sites and focus on the other five acetylation sites and K497.

To enable detailed and more informative kinetic studies, we expressed in *Escherichia coli* (*E. coli*) and purified the six site-specifically acetylated G6PD variants mentioned above, as well as WT

and AcK414 as controls (Fig. 2a and Supplementary Fig. 3). Next, we measured steady-state kinetic parameters using equal amounts of purified G6PD. Initial velocity versus G6P or $NADP^+$ concentration curves were measured by fixing the other substrate at a saturating concentration (Supplementary Fig. 4a–c). Apparent $V_{max}$ ($V_{max}^{app}$) and $K_M$ ($K_M^{app}$) values were calculated by fitting the data to the Michaelis–Menten equation (Fig. 2b and Supplementary Table 2).

The kinetic parameters of WT G6PD measured for the substrate G6P were in agreement with previously reported values[48]. In addition, $V_{max}^{app}$ measured for recombinant pWT was comparable to $V_{max}^{app}$ of recombinant WT (Supplementary Fig. 4d), validating its use as a non-acetylated control in catalytic activity measurements performed in cell lysates (hereafter, we refer to AcK414 as pWT G6PD). According to the measured kinetic parameters (Supplementary Table 2), AcK386, AcK408, AcK432, and AcK497 G6PD had 30–70% lower enzymatic efficiency ($K_{cat}^{app}/K_M^{app}$) compared to WT G6PD, AcK403 displayed significantly lower activity, and AcK89 G6PD displayed similar $K_{cat}^{app}/K_M^{app}$

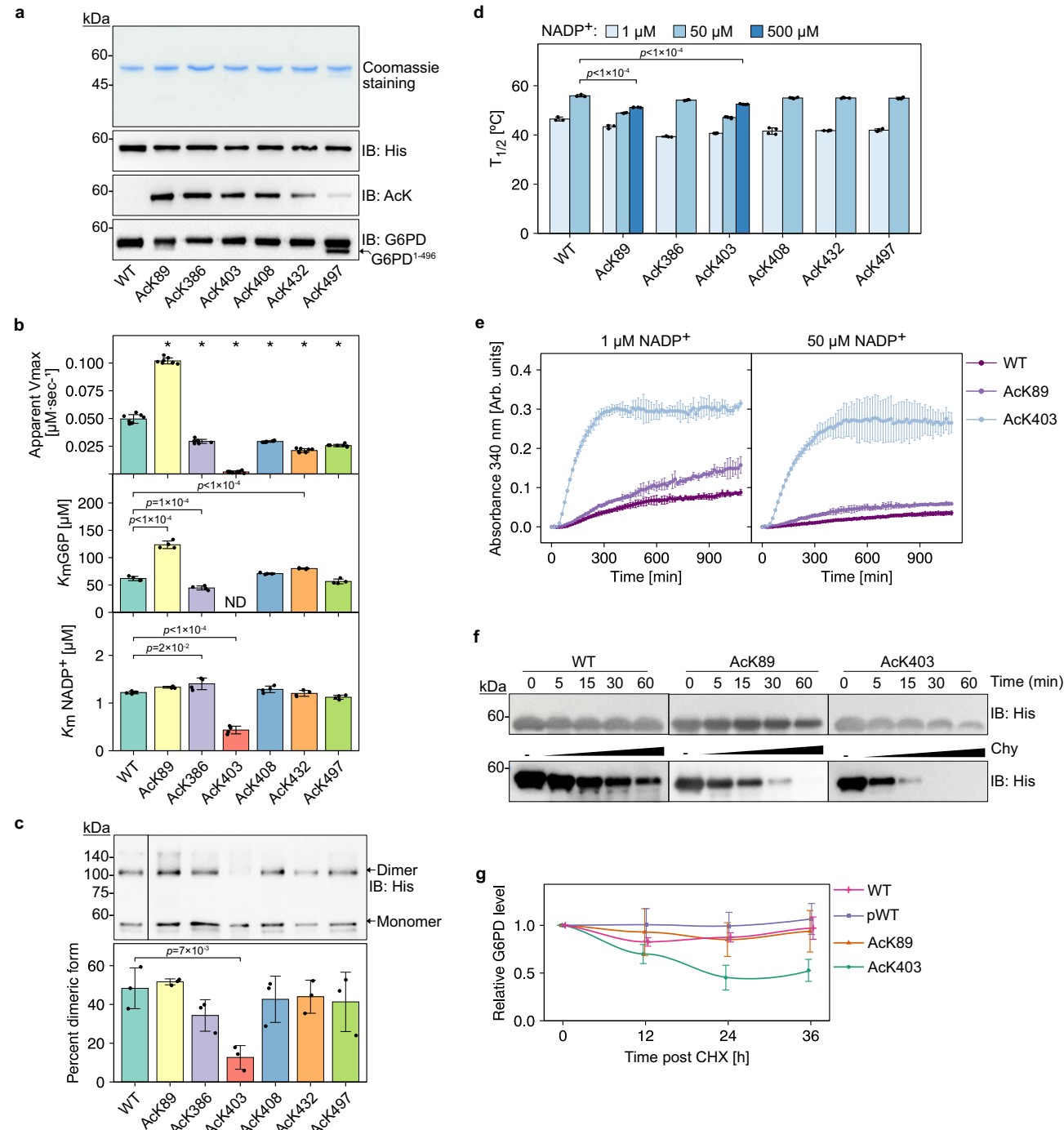

**Fig. 2 | Lysine 403 acetylation inhibits and destabilizes G6PD in vitro and in cells. a** Purified site-specifically acetylated full-length G6PD expressed in *E. coli*. **b** Site-specific acetylation significantly affects the catalytic activity of recombinant G6PD. Bars represent the mean apparent kinetic parameters (obtained by fitting the data to the Michaelis−Menten equation) ± fitting error; $n = 8$ ($V_{max}$), or 4 ($K_M^{app}$) independent experiments (Supplementary Fig. 4a, b). Data were analyzed using one-way ANOVA followed by Tukey's post hoc test. * $p < 1 \times 10^{-4}$. **c** Acetylation of K403 destabilizes the dimeric structure of G6PD. Indicated purified G6PD variants were cross-linked, resolved by SDS-PAGE, and visualized by immunoblotting using an antibody against the C-terminal 6×His tag. Bars represent the percentage of dimeric G6PD out of the total G6PD population [monomeric (Mr: ~59 KDa), and dimeric (Mr: ~120 KDa)]. Data were analyzed using one-way ANOVA followed by Tukey's post hoc test and are presented as mean values ± SD; $n = 3$ independent experiments. **d** Lysine 403 acetylation reduces the thermodynamic stability of G6PD. Bars represent mean $T_{1/2}$ values ± SD, measured at increasing

concentrations of NADP+ (Supplementary Fig. 5); $n = 3$ (WT, AcK89, AcK386), or 4 (AcK403, AcK408, AcK432, AcK497). Data were analyzed using one-way ANOVA followed by Tukey's post hoc test. **e** Lysine 403 acetylation increases the in vitro aggregation propensity of G6PD. Curves represent the accumulation of G6PD aggregates as a function of time and indicated concentration of NADP+. Aggregation was monitored by measuring the increase in scattered 340 nm light. Data are the mean ± SD, $n = 3$ independent experiments. **f** Lysine 403 acetylation increases the proteolytic degradation rate of G6PD in vitro. Western blots show the amount of indicated G6PD variants following chymotrypsin (Chy) digestion as a function of time (top) or as a function of chymotrypsin concentration (bottom). **g** Lysine 403 acetylation destabilizes G6PD in cells. Curves represent the relative amounts of G6PD variants in HCT116 cells as a function of time after CHX treatment. Data are the mean ± SD, $n = 3$ (WT, AcK403) or 5 (pWT, AcK89) biologically independent samples. ND not determined. Source data are provided as a Source Data file.

for G6P, but 2-fold higher catalytic efficiency for NADP$^+$. Thus, using site-specifically acetylated G6PD, we show that G6PD can be activated and not only deactivated by the acetylation of a single lysine residue.

Oligomerization is essential for G6PD activity, and the majority of natural G6PD mutations leading to severe G6PD deficiency are clustered around the structural NADP$^+$ binding site, affecting G6PD oligomerization, stability, and activity[38,47,49–51]. To determine if the measured acetylation-dependent variations in G6PD activity correlate with G6PD oligomerization, we evaluated the oligomeric state of recombinant G6PD variants by in vitro crosslinking using disuccinimidyl suberate (DSS). We confirmed that the dimeric population of AcK403 G6PD was significantly smaller compared to WT and other acetylated G6PD variants (Fig. 2c)[34], in line with its impaired enzymatic activity, and the position of K403 within the structural NADP$^+$ binding site, close to the dimer interface. In contrast, acetylation of other lysine residues had no significant effect on G6PD dimerization.

In addition, we evaluated the thermodynamic stability of acetylated G6PD by measuring residual enzymatic activity following heat denaturation (Fig. 2d and Supplementary Fig. 5). In the presence of 1 µM NADP$^+$, $T_{1/2}$ of acetylated G6PD variants was lower by 3.4–7.6 °C, compared to WT G6PD. The thermal stability of all enzymes increased upon addition of 50 µM NADP$^+$, but even in the presence of 500 µM NADP$^+$, the $T_{1/2}$ of AcK89 and AcK403 G6PD was significantly lower than the $T_{1/2}$ of WT G6PD at 50 µM NADP$^+$. Collectively, these data show that the acetylation of K89 and K403 lowers the thermodynamic stability of G6PD, while K403 acetylation also negatively affects its dimerization. Following these findings, we continued our study with K89 and K403 acetylation sites that activate and deactivate G6PD, respectively.

To further characterize the stability of K89- and K403-acetylated G6PD, we followed the aggregation of G6PD over time (Fig. 2e). Both variants displayed higher aggregation propensity, but AcK403 displayed faster and more robust aggregation even at 50 µM NADP$^+$. Furthermore, AcK403 was more susceptible to chymotrypsin digestion than WT G6PD (Fig. 2f), in line with its monomeric structure and lower thermodynamic stability. We also determined the effect of acetylation on G6PD stability in cells using cycloheximide (CHX; Fig. 2g). In agreement with in vitro measurements, AcK403 was degraded faster, reaching ~55% of its initial level 24 h post CHX addition. Thus, our data show that K89 and K403 acetylation affect G6PD stability; in particular, K403 acetylation markedly reduced the thermodynamic stability of human G6PD, with a clear indication of impaired dimerization, increased aggregation propensity, and shorter half-life in cells.

## K89 acetylation promotes local conformational changes and ubiquitylation of adjacent K95/K97

Proteins can be posttranslationally modified by multiple PTMs at multiple sites simultaneously, and one PTM may affect the addition, removal, or function of other PTMs. For example, acetylation and ubiquitylation often compete for the same lysine residues on a protein, thereby regulating its function and stability. In G6PD, K95 and K97 were found to be both acetylated and ubiquitylated[22]. While our data show that K95 and K97 acetylation had no effect on catalytic activity, we asked if K89 acetylation modulates the ubiquitylation of nearby K95 and K97. Western blot analysis found that the ubiquitylation level of AcK89 G6PD was higher, relative to pWT, AcK95, and AcK97 G6PD, suggesting that K89 acetylation promotes the ubiquitylation of G6PD (Fig. 3a). Next, we introduced K95R, K97R, or K95/97R mutations into pWT and AcK89 G6PD and verified that these mutations have no effect on $V_{max}$ (Fig. 3b). Mutation of either K95 or K97 to arginine reduced the level of AcK89 G6PD ubiquitylation considerably (Fig. 3c), showing that K89 acetylation promotes the ubiquitylation of G6PD on residues K95/97.

Crosstalk between PTMs can be attributed to PTM-dependent structural, conformational, and electrostatic effects. To test the hypothesis that K95/97 ubiquitylation is affected by K89 acetylation-dependent structural changes, we determined the crystal structure of K89-acetylated G6PD. AcK89 G6PD was crystallized in the F222 space group with one protein molecule and structural NADP$^+$ within the asymmetric unit, and the structure was solved to a resolution of 2.46 Å (statistics of data collection and model refinement are listed in Supplementary Table 3). Superposition of monomeric AcK89 and WT G6PD (PDB ID: 6E08)[47] revealed a Cα root mean square deviation (RMSD) value of 0.24 Å, suggesting only minor effects of K89 acetylation on G6PD backbone structure (Fig. 3d). Similarly, the dimeric structures of AcK89 (obtained by symmetry operations) and WT G6PD were essentially identical (C$_\alpha$ RMSD value of 0.35 Å; Supplementary Fig. 6a). In contrast, notable local conformational changes were found around residue AcK89, which was rotated by ~180° relative to non-acetylated K89 (Fig. 3e). This conformational change resulted in the partial unwinding of helix αb' and rotation of residues E94−K97 by ~45° relative to the helix axis (Fig. 3f and Supplementary Movie 1). Thus, according to the crystal structure of AcK89, ubiquitylation of K95/97 may be promoted by K89 acetylation-dependent local conformational changes.

## AcK403 G6PD is reactivated by Sirt1- and Sirt2-dependent deacetylation

Lysine acetylation is a highly dynamic PTM, and its levels are determined by the opposing activities of KATs and KDACs. Specifically, K403 can be acetylated by KAT9[34], and we show here that it is also a substrate of CREB binding protein (CBP; Supplementary Fig. 6b−d). In addition, it was found that K403 acetylation is regulated by Sirt2 but not Sirt1[34]. To study the deacetylation of G6PD in a non-biased way (i.e., using site-specifically acetylated substrates), we measured the expected change in G6PD catalytic activity upon deacetylation (Fig. 4a). In agreement with our previous measurements (Fig. 1d), AcK171 was catalytically inactive, independent of KDACi treatment. The catalytic activity of pWT was not affected by KDACi, while endogenous G6PD was inhibited by KDACi (Fig. 1a; Fig. 4a). This can be explained by the higher abundance of pWT, leading to low acetylation stoichiometry due to insufficient acetylation by endogenous KATs[52]. In contrast to AcK171 and pWT, the activating effect of K89 acetylation was maintained only when AcK89 G6PD was expressed in cells cultured with both NAM and SAHA. In addition, AcK403 G6PD displayed catalytic activity comparable to pWT G6PD when expressed without KDACi or with SAHA only, but its activity was reduced to ~40% upon incubation with both inhibitors or NAM only (Fig. 4a). Similarly, immunoblot measurements of immunoprecipitated G6PD found higher levels of K89 acetylation when cells were cultured in the presence of both inhibitors, and higher levels of K403 acetylation following incubation with NAM or NAM and SAHA (Fig. 4b).

Next, we followed the deacetylation of AcK89 and AcK403 in vitro. Incubation of recombinant AcK89 G6PD with cleared lysate of naïve HEK293T cells cultured with or without KDACi did not affect G6PD activity, suggesting higher stability of K89 acetylation towards in vitro deacetylation (Fig. 4c). However, in agreement with activity measurements in cell lysates, incubating recombinant AcK403 with lysate of cells cultured without KDACi or with SAHA only, resulted in the recovery of G6PD activity, indicative of AcK403 deacetylation. As expected, pretreatment of cells with NAM or both NAM and SAHA resulted in a negligible recovery of catalytic activity (Fig. 4c). Collectively, these data show that AcK403 is mainly deacetylated by NAD$^+$-dependent deacetylases and is more labile than AcK89.

To identify specific deacetylases involved in regulating K89 and K403 acetylation state, we performed an in vitro deacetylation assay by incubating recombinant acetylated G6PD with immunoprecipitated Sirt1 and Sirt2, which, like G6PD, are localized in the cytosol. We also

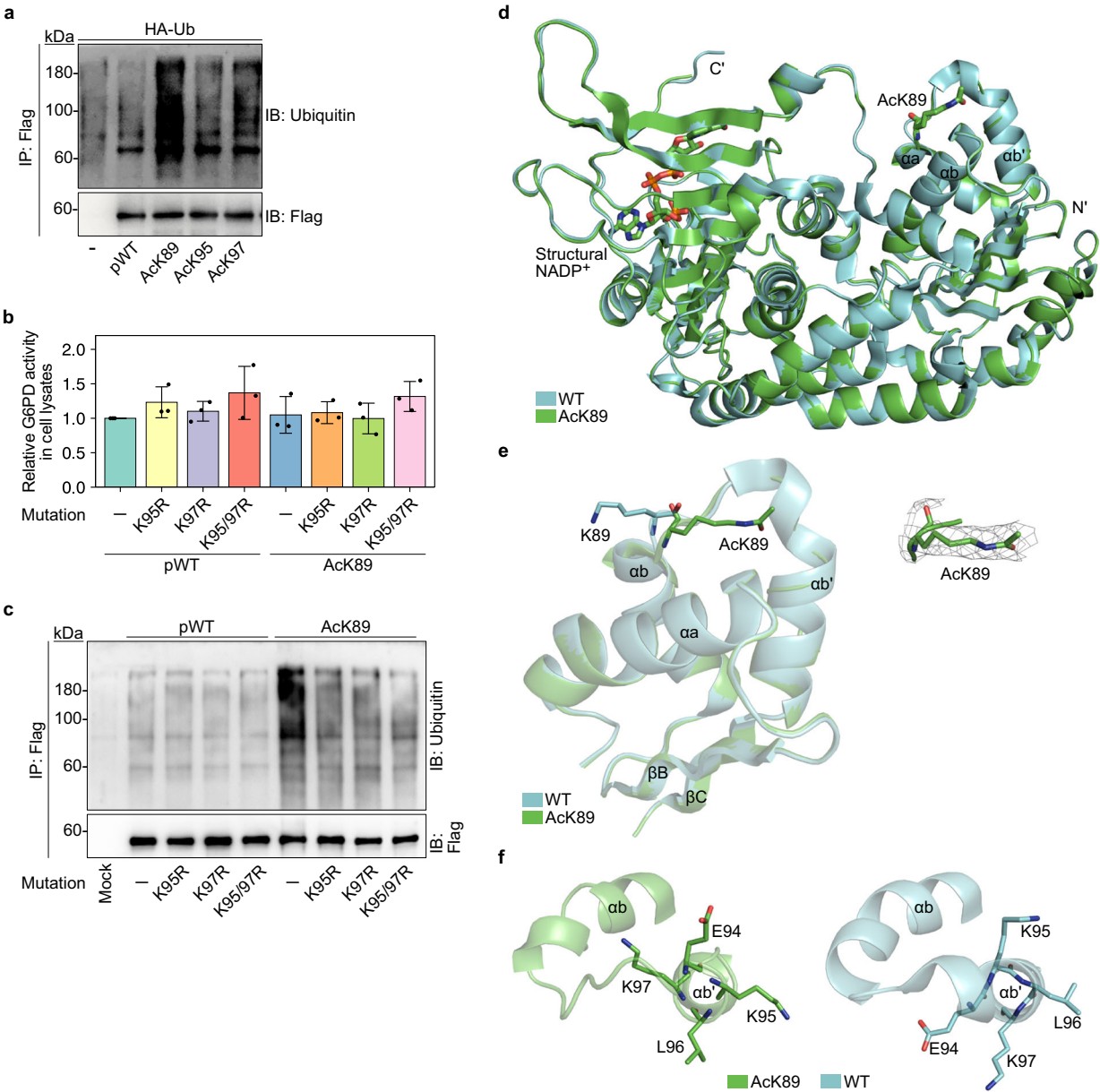

**Fig. 3 | Lysine 89 acetylation promotes partial unwinding of helix αb' and ubiquitylation of residues K95/97. a** Lysine 89 acetylation promotes ubiquitylation of G6PD. Representative Western blots showing the ubiquitylation level of immunoprecipitated Flag-tagged G6PD variants, co-expressed with HA-tagged ubiquitin in HEK293T cells treated with 5 μM MG132 for 16 h. **b** Mutation of lysine 95 and 97 to arginine does not affect G6PD activity. Bars represent the relative $V_{max}$ of pWT, AcK89, and indicated Lys-to-Arg G6PD mutants, measured in lysates of cells incubated in the absence of KDACi, and displayed relative to pWT G6PD. Data are the mean ± SD, $n = 3$ biologically independent samples. **c** AcK89 G6PD is ubiquitylated on residues K95/97. Representative Western blots showing the ubiquitylation level of immunoprecipitated Flag-tagged G6PD variants, co-expressed with HA-tagged ubiquitin in HEK293T cells treated with 5 μM MG132 for 16 h. **d** Lysine 89 acetylation has minimal effect on the three-dimensional structure of monomeric G6PD. Superposition of AcK89 G6PD (green) and non-acetylated G6PD (cyan, PDB ID: 6E08). Structural NADP[+] and K89 are shown in sticks model. **e** Lysine 89 acetylation promotes local conformational changes. Left: zoom in on K89 and nearby helices, showing the position of AcK89 (green) relative to the position of K89 (cyan, PDB ID: 6E08). Right: electron density map around residue AcK89. The $2F_o-F_c$ map was contoured at 1 σ. **f** Lysine 89 acetylation promotes partial unwinding of helix αb'. Local conformational changes around AcK89 with the partial unwinding of helix αb' (green), relative to the structure of non-acetylated G6PD (cyan, PDB ID: 6E08). Source data are provided as a Source Data file.

included HDAC6, which was found to directly interact with G6PD (Supplementary Fig. 6e, f)[53]. While K89 remained acetylated under these conditions, AcK403 G6PD was efficiently deacetylated in vitro by Sirt1 and Sirt2 (but not HDAC6), and NAM blocked its deacetylation (Fig. 4d). Similarly, incubation with immunoprecipitated Sirt1 and Sirt2, but not HDAC6, restored the catalytic activity of recombinant AcK403 G6PD and NAM inhibited the reactivation of AcK403 by both enzymes (Fig. 4e). Deacetylation measurements as a function of time confirmed that AcK403 is a substrate of Sirt1 and Sirt2 (Fig. 4f). In light

of the relative stability of AcK89 towards in vitro deacetylation, we co-expressed AcK89 G6PD with selected deacetylases and found that Sirt1 and Sirt2 can catalyze the hydrolysis of AcK89, but only when over-expressed in cultured cells (Supplementary Fig. 6g). We noticed an apparent decrease in AcK89 acetylation upon co-expression with HDAC6, which can be explained by HDAC6-dependent reduced amber suppression efficiency[54]. Taken together, our data show that K403 acetylation is more labile than K89 acetylation and that AcK403 G6PD can be re-activated by Sirt1- and Sirt2-dependent deacetylation.

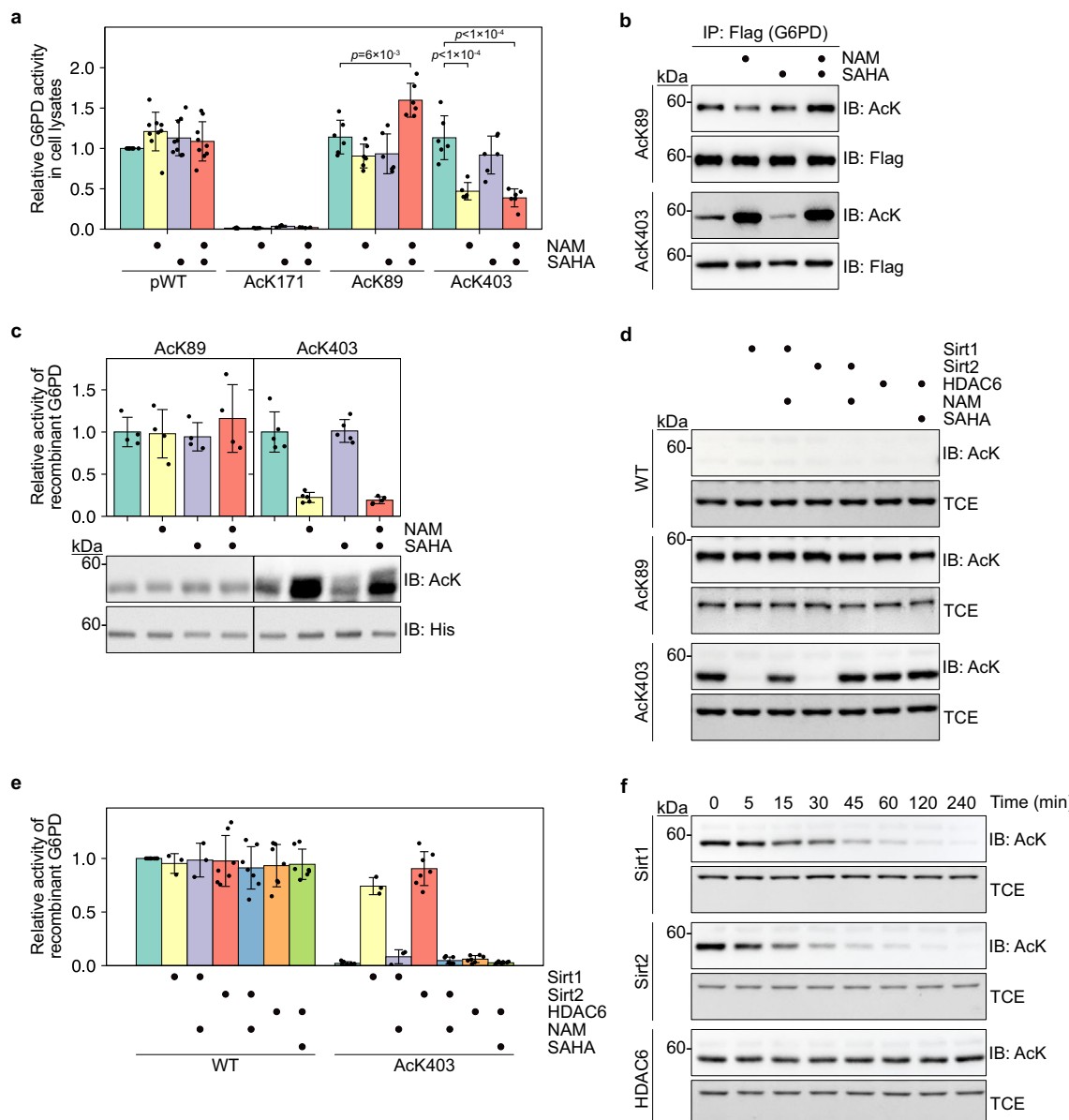

**Fig. 4 | Lysine 403 acetylation is regulated by Sirt1 and Sirt2. a** The effect of acetylation on G6PD activity is maintained following the inhibition of endogenous KDACs. Bars represent relative $V_{max}$ of acetylated G6PD variants expressed in HEK293T cells cultured in the presence or absence of indicated KDACi. The activity was measured in cleared total cell lysates, corrected to G6PD expression level, and normalized to the activity of pWT expressed without KDACi treatment. Data were analyzed using one-way ANOVA followed by Tukey's post hoc test and are presented as mean values ± SD; $n = 4$ (AcK171), 5 or 6 (AcK403), 6 (AcK89), or 9 (pWT) biologically independent samples. **b** Lysine 403 acetylation is stabilized upon inhibition of endogenous NAD⁺-dependent deacetylases. Representative Western blots showing the acetylation level of immunoprecipitated AcK89 and AcK403 G6PD, expressed in HEK293T cells cultured in the presence or absence of indicated KDACi. **c** The inhibitory effect of K403 acetylation is alleviated by NAD⁺-dependent deacetylases in vitro. Bars represent the relative $V_{max}$ of recombinant G6PD

incubated with cleared lysates of HEK293T cells pretreated with indicated KDACi [mean ± SD; $n = 4$ (AcK89), or 5 (AcK403) independent experiments]. Western blots show the amount of G6PD (anti 6×His) and acetylation level following incubation with cleared cell lysates. **d** Lysine 403-acetylated G6PD is a substrate of Sirt1 and Sirt2. Western blots show the acetylation level of recombinant WT, AcK89, and AcK403 G6PD, following incubation with immunoprecipitated Sirt1, Sirt2, or HDAC6, in the presence or absence of indicated KDACi. **e** Sirt1 and Sirt2 reactivate AcK403 G6PD. Bars represent the relative $V_{max}$ of recombinant WT and AcK403 G6PD, following incubation with deacetylases as described in **d**. Data are the mean ± SD; $n = 3$ (Sirt1 treatment), or 7 (Sirt2, HDAC6 treatment) independent experiments. **f** Sirt1 and Sirt2 deacetylate AcK403 in a time-dependent manner. Representative Western blots showing K403 acetylation level as a function of incubation time with immunoprecipitated Sirt1, Sirt2, or HDAC6. Source data are provided as a Source Data file.

## K403 acetylation inhibits G6PD activity via long-range conformational changes

To elucidate the molecular mechanism of G6PD inactivation by KAT9- and CBP-dependent K403 acetylation, we determined the crystal structure of AcK403 G6PD. The protein crystallized in the P1 space group, with eight monomers within the asymmetric unit, and the structure was determined at 2.28 Å resolution (Supplementary

Table 3). The Cα RMSD for the superposition of monomeric AcK403 and WT G6PD (PDB ID: 6E08) was 0.8 Å, indicating that the overall backbone structure of the two monomers is similar (Supplementary Fig. 7a). That said, the structure of AcK403 included an α helix at the N-terminus (residues R9–Q22, hereafter referred to as αa') that has not been observed in other crystal structures of human G6PD. The dimeric structure of AcK403 G6PD was assembled by symmetry operations

(Fig. 5a). Comparison to WT G6PD dimer by superimposing one monomer (Fig. 5b, light shades) and using Glu93-C$\alpha$ of the other monomer as a reference point, found a large effect of AcK403 acetylation on the relative orientation of G6PD monomers in the dimeric structure, in line with the observed difference in the oligomeric state of AcK403.

Lysine 403-acetylated G6PD was crystallized without structural or catalytic NADP$^+$, and a close examination of the structure revealed acetylation-dependent conformational changes at the dimer interface and the substrate binding site. In non-acetylated G6PD, positively charged K403 and R370 interact with the phosphate groups of NADP$^+$. Following K403 acetylation, the acetylated residue adopts a different conformation that is stabilized by electrostatic interaction with R370, excluding the structural NADP$^+$ from its binding site (Fig. 5c, and Supplementary Fig. 7b, c). This minor conformational change in the

structural NADP$^+$ binding site was accompanied by distortion of the $\beta$-sheet at the dimer interface (mainly $\beta$N) and unwinding of the short $\alpha$l helix (Fig. 5c and Supplementary Fig. 7d). The unwinding of $\alpha$l was accompanied by additional, long-range structural deformations. First, the newly formed loop (formerly $\alpha$l) reshaped the dimer interface (Fig. 5c), in line with the observed differences in the dimeric structure (Fig. 5b). Although the K403Q mutation can mimic the inhibitory effect of K403 acetylation, the dimeric structure of AcK403 was substantially different from that of K403Q G6PD (PDB ID: 7SEI; Supplementary Fig. 7e, f)[38]. Second, in the structure of AcK403 G6PD, the unwinding of $\alpha$l promoted the formation of $\beta$-sheet-like interactions with the N terminus of the protein (residues V5–L7, Fig. 5d). These stabilizing backbone interactions probably enabled the identification of the additional $\alpha\alpha'$ helix, which is involved in hydrophobic interactions with helix $\alpha$m (Fig. 5d). Third, in AcK403 G6PD, K205 was rotated by ~180°

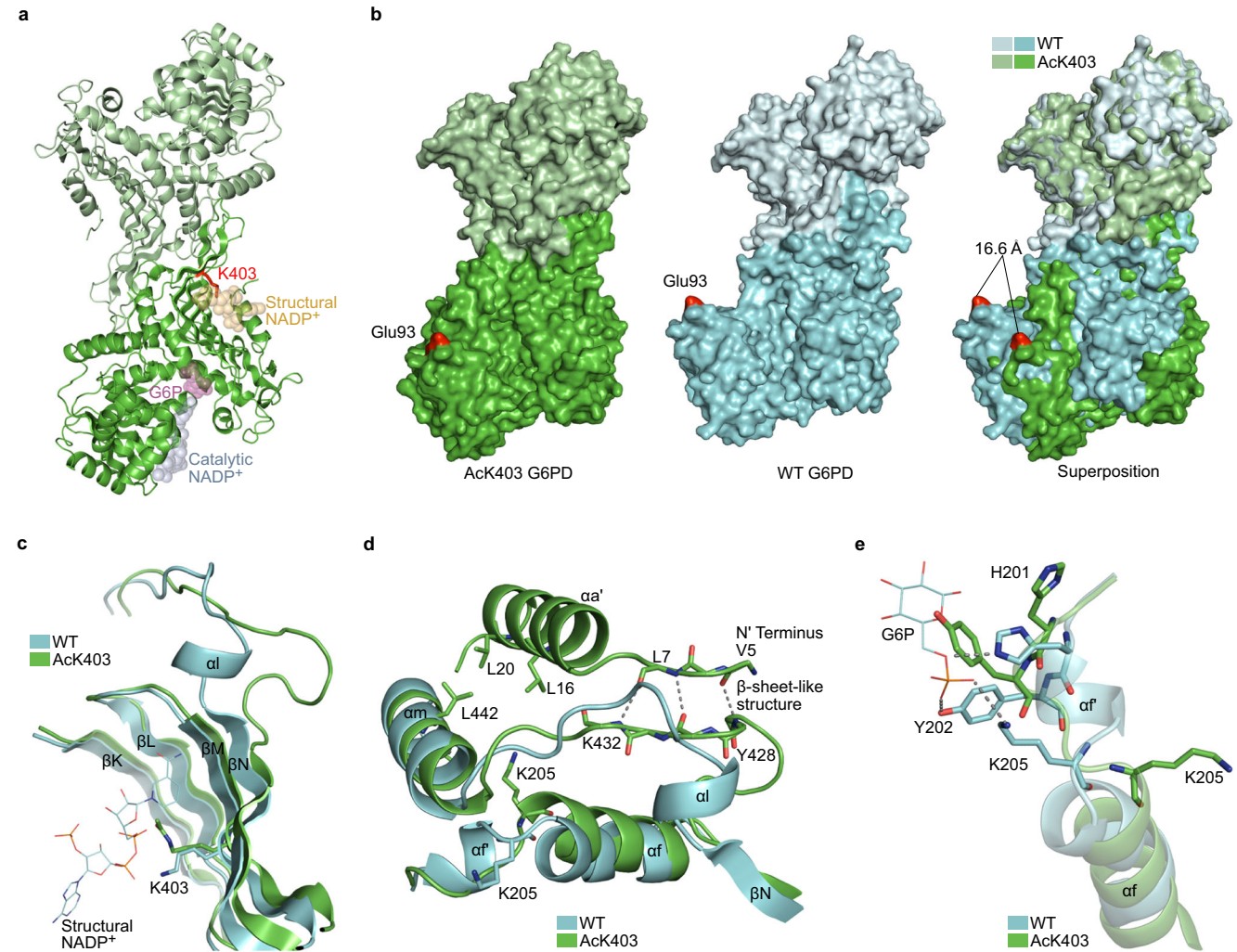

**Fig. 5 | Lysine 403 acetylation promotes structural changes at the dimer interface and the active site. a** Dimeric structure of AcK403 G6PD. The expected position of catalytic NADP$^+$ (light blue), structural NADP$^+$ (light orange), and G6P (light purple) are presented as transparent spheres, based on superposition of AcK403 G6PD structure with structures of WT and mutant G6PD (PDB ID: 6VA7, 6E08, and 2BHL)[47,49,73]. Acetylated lysine residue 403 is highlighted in red. **b** Lysine 403 acetylation affects the dimeric structure of G6PD. Space-filling model of dimeric AcK403 G6PD (left) and WT G6PD (center, PDB ID: 6E08), with residue Glu93 highlighted in red. Right: The orientation of the lower monomers (darker shades) relative to the upper monomers (lighter shades) is compared by superimposing the upper monomers of each dimeric structure. The distance between C$\alpha$ atoms of Glu93 residues is shown. **c** Lysine 403 acetylation is accompanied by

distortion of $\beta$N and unwinding of $\alpha$l. Superposition of AcK403 G6PD (green) and WT G6PD (cyan, PDB ID: 6E08), with a close-up view of the structural NADP$^+$ binding site. Lysine 403 is presented in sticks model. **d** Close-up view of helices $\alpha$f and $\alpha\alpha'$ in AcK403 G6PD (green) and WT G6PD (cyan, PDB ID: 6E08). Backbone hydrogen bonds between residues V5–L7 and Y428–K432 that form a $\beta$-sheet-like structure are shown in gray. Residues L16, L20, and L442 that form a hydrophobic core between helices $\alpha\alpha'$ and $\alpha$m are rendered in sticks model. Also shown is residue K205, which rotates by ~180° following K403 acetylation and unfolding of helix $\alpha$f'. **e** Lysine 403 acetylation promotes long-range conformational changes in the active site. Close-up view of the G6P binding site in AcK403 G6PD (green) and WT G6PD (cyan, PDB ID: 2BHL). The position of the substrate G6P in WT G6PD is shown, together with polar interactions with residues H201, Y202, and K205 (gray).

relative to its position in WT G6PD, and held in position by electrostatic interaction with D443 (Fig. 5d, and Supplementary Fig. 7g, h). This conformational change was accompanied by the unwinding of helix αf', the disposition of helix αf away from the substrate binding site, and the reorientation of functionally important residues such as Y202, and H201, that together rendered the enzyme inactive (Fig. 5e and Supplementary Fig. 7i). In summary, K403 acetylation-dependent conformational changes in the NADP⁺ binding site reshaped the dimer interface and propagated to the substrate binding site via partial unwinding of helices αl and αf', disposition of helix αf, and deformation of the G6P binding site (Supplementary Movie 1).

## K403 acetylation promotes Fyn-dependent phosphorylation of Y503

Considering the structural changes at the dimer interface and the structural NADP⁺ binding site, we hypothesized that K403 acetylation might affect the phosphorylation of nearby Y401 by Fyn, a non-receptor tyrosine kinase and a member of the Src family of kinases[55]. To check our hypothesis, we first confirmed by LC-MS/MS analyses that pWT G6PD is phosphorylated on Y401 when co-expressed with Fyn and not with the dominant negative mutant of Fyn (FynDN; Supplementary Fig. 8). A peptide fragment of AcK403 G6PD that includes positions Y401 and AcK403 was not identified in our LC-MS/MS analyses; therefore, the phosphorylation status of Y401 in AcK403 G6PD could not be determined. However, we found that AcK403 G6PD, and not pWT G6PD, was phosphorylated on Y503. To confirm these results, we introduced Y401F and Y503F mutations into WT, pWT, and AcK403 G6PD and verified that these mutations have no effect on $V_{max}$ of G6PD expressed both in bacteria and in HEK293T cells (Supplementary Fig. 9a–c). In agreement with MS data, Western blot analysis found WT and pWT G6PD to be phosphorylated when co-expressed with Fyn and not FynDN, and Y401F mutation significantly reduced their phosphorylation (Fig. 6a and Supplementary Fig. 9d). Moreover, Y503F mutation had no effect on pWT phosphorylation, indicating that Y503 is not a major Fyn-phosphrylation site in non-acetylated G6PD. In contrast, in AcK403 G6PD, Y503F mutation, but not Y401F, significantly reduced Fyn-dependent AcK403 G6PD phosphorylation in cells (Fig. 6a). These data show that Fyn-dependent Y503 phosphorylation is promoted by K403 acetylation. According to our structure of AcK403 G6PD, Y503 is positioned close to the structural NADP⁺ binding site, ~13.6 Å from AcK403, which may explain the crosstalk between these PTMs.

To determine whether Fyn directly phosphorylates G6PD, we carried out a non-radioactive in vitro kinase assay (Supplementary Fig. 9e). In agreement with Fyn activity in cultured cells, we found that in WT G6PD, Y401 is a major Fyn phosphorylation site and that Y503 is not a substrate of Fyn (Fig. 6b). However, in AcK403 G6PD, Y401F and Y503F mutations significantly reduced Fyn-dependent AcK403 phosphorylation in vitro, demonstrating that Fyn directly phosphorylates Y503 in a K403 acetylation-dependent manner. Despite the close distance between K403 and Y503, phosphorylation of the latter had no effect on the deacetylation of AcK403 by Sirt1 and Sirt2 (Supplementary Fig. 9f,g). Furthermore, in contrast to studies in red blood cells[55], we found no significant effect of Fyn phosphorylation on $V_{max}$ of G6PD (Supplementary Fig. 9h–k). Collectively, these data show that G6PD Y503 is a bona fide Fyn phosphorylation site and that the acetylation of G6PD on K403 promotes Y503 phosphorylation by Fyn.

## K403 acetylation modulates Fyn-dependent phosphorylation via interaction with p53

Our data show that K403 acetylation deforms the dimer interface and renders G6PD monomeric. According to Jiang et al.[56], G6PD can bind p53, and this interaction prevents G6PD dimerization, suggesting binding of p53 to areas in G6PD involved in or affected by dimer formation. Therefore, we asked if the binding of p53 prevents G6PD dimerization, or whether p53 has a higher affinity to monomeric G6PD.

To answer this question, we performed pull-down experiments using purified 6×His-tagged G6PD and HA-tagged p53 expressed in HEK293T cells, and found that the interaction between G6PD and p53 is driven by K403 acetylation (Fig. 6c). Moreover, although the acetylation-mimic K403Q mutation renders G6PD more monomeric, it did not stabilize the interaction between G6PD and p53, suggesting that this interaction is dependent on local interactions and not only the oligomeric state of G6PD (Supplementary Fig. 10a). We also evaluated G6PD-p53 interaction indirectly, by chasing p53 degradation in HCT116 cells. In cells expressing AcK150 GFP, pWT, AcK89, or AcK171 (as controls), the amount of p53 was reduced by half within 2 h after CHX treatment (Fig. 6d and Supplementary Fig. 10b). However, the amount of p53 in cells expressing AcK403 G6PD was ~20% higher. Thus, K403 acetylation promotes the interaction between p53 and G6PD and stabilizes p53 in cells.

The effect of K403 acetylation on G6PD oligomerization, Fyn-dependent phosphorylation, and interaction with p53 implies that Fyn and p53 may recognize similar or close positions in G6PD. Pull-down of 6×His-tagged pWT or AcK403 G6PD, co-expressed with Fyn in E. coli, found a direct interaction between G6PD and Fyn (Supplementary Fig. 10c). Importantly, K403 acetylation stabilized the interaction between G6PD and Fyn, similar to its effect on G6PD-p53 interaction. Therefore, we asked if the observed G6PD-p53 interaction can modulate G6PD phosphorylation by Fyn. Since p53 and G6PD migrate at similar sizes in sodium dodecyl sulfate-polyacrylamide gel electrophoresis (SDS-PAGE), we chose an N'-truncated p53 variant, lacking the first 66 residues (Δ66 p53, hereafter referred to as p53')[57], which showed interaction with WT and AcK403 G6PD comparable to full-length p53 (Supplementary Fig. 10d). p53 is not a known substrate of Fyn. However, we found that Fyn can phosphorylate p53 in vitro (Fig. 6e and Supplementary Fig. 10e). Moreover, phosphorylation of WT and AcK403 G6PD by Fyn decreased as the molar ratio between p53' and G6PD changed from 1:16 to 1:1 (p53:G6PD; Supplementary Fig. 10e). Importantly, incubation with substoichiometric amounts of p53' (p53:G6PD molar ratio of 1:16–1:8) reduced WT G6PD phosphorylation by 20%–30% but did not affect AcK403 phosphorylation, although the interaction between AcK403 G6PD and p53 was not blocked by Fyn phosphorylation (Fig. 6f, left graph, and Supplementary Fig. 10f). In excellent agreement, Fyn-dependent p53' phosphorylation was lower in the presence of AcK403 G6PD but not WT G6PD (Fig. 6f, right graph). Higher concentration of p53' (p53:G6PD molar ratio of 1:4–1:1), significantly reduced the phosphorylation level of both WT and AcK403 G6PD (Fig. 6g and Supplementary Fig. 10e). Collectively, these data show that p53' interferes with Fyn-dependent G6PD phosphorylation and that at substoichiometric amounts of p53', K403 acetylation can partially counteract the inhibitory effect of p53'.

## K403 acetylation promotes the induction of apoptotic signaling

p53 plays an important role in cell metabolism and can also mediate apoptosis by direct activation of Bax[58]. Therefore, we asked if the acetylation-dependent p53:G6PD interaction and stabilization of p53 have a role in p53-regulated cellular processes. Indeed, we noticed an increase in levels of the pro-apoptotic protein Bax, in HEK293T cells expressing AcK403 G6PD but not pWT G6PD (Fig. 6h). Unexpectedly, the accumulation of Bax was observed mainly in cells incubated in the absence of NAM, although K403 acetylation level is very low under these conditions. Moreover, we found a similar increase in Bax levels in p53 WT HCT116 (HCT⁺/⁺) cells but not in p53 knockout HCT116 (HCT⁻/⁻) cells (Supplementary Fig. 11a), suggesting that G6PD K403 acetylation promotes the accumulation of pro-apoptotic Bax in a p53-dependent manner. Together with the increase in Bax levels, we found a decrease in cellular levels of Bcl2 and an increase in cleaved caspase-3 levels (Fig. 6i and Supplementary Fig. 11b). These effects were not observed in HEK203T or WT HCT116 cells expressing the acetylation-resistant K403R G6PD mutant nor the acetyl-mimic K403Q mutant

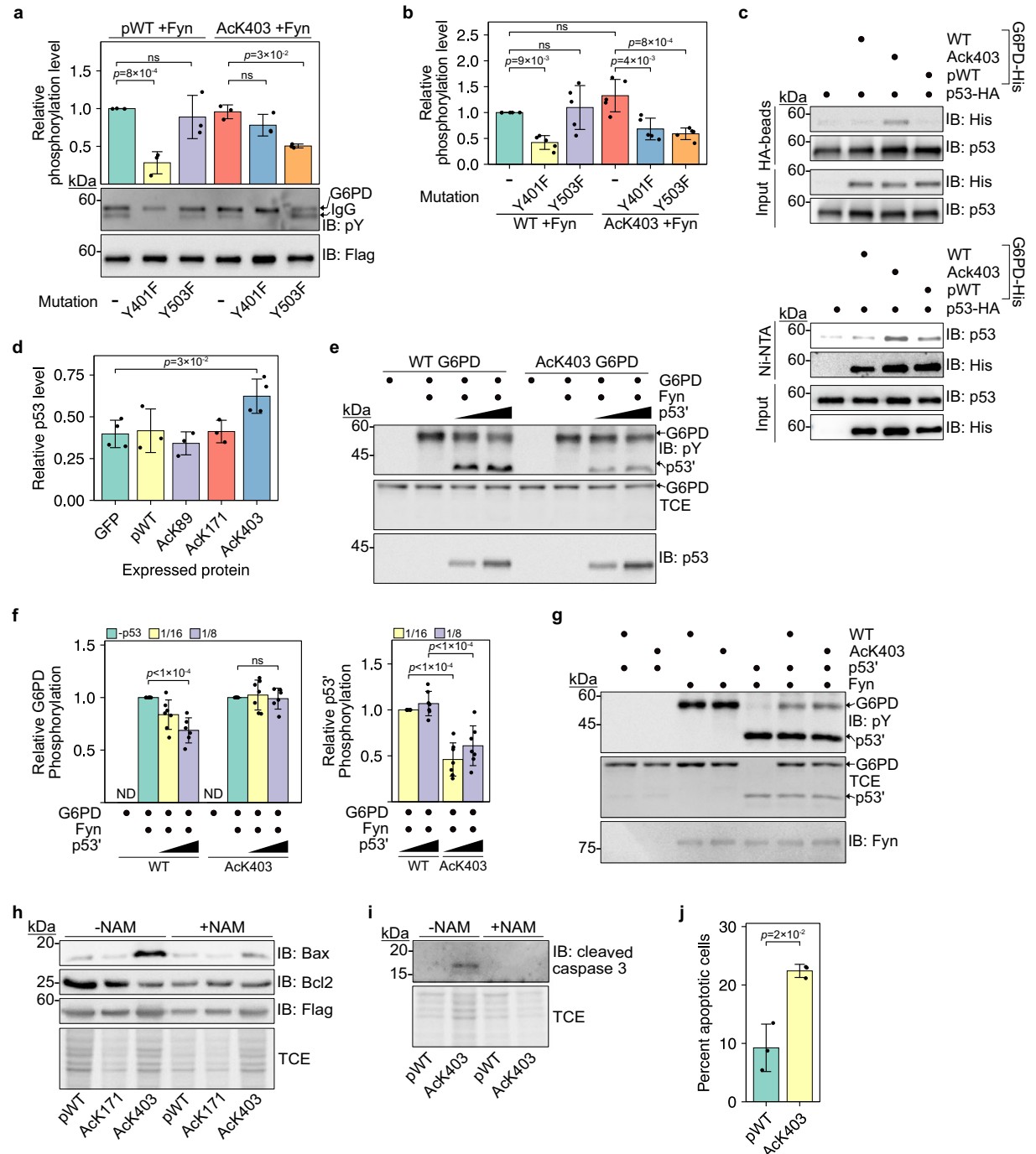

**Fig. 6 | p53 modulates Fyn-dependent phosphorylation as a function of K403 acetylation. a** Y401 and Y503 are major Fyn phosphorylation sites in pWT and AcK403 G6PD, respectively. Representative Western blots showing tyrosine phosphorylation of immunoprecipitated G6PD co-expressed with Fyn. Data were analyzed using one-way ANOVA followed by Tukey's post hoc test and presented as mean values ± SD; $n = 3$ biologically independent samples. **b** G6PD is phosphorylated by Fyn in vitro. Bars represent the relative phosphorylation level of recombinant G6PD, following in vitro incubation with Fyn. Data were analyzed using one-way ANOVA followed by Tukey's post hoc test and are presented as mean values ± SD; $n = 5$ independent experiments. **c** Acetylation of G6PD K403 promotes the interaction with p53. Top: Co-IP using HA-tagged p53. Bottom: Reciprocal binding using 6×His-tagged G6PD. **d** AcK403 G6PD stabilizes p53 in cells. Bars represent the relative immunoblot intensities of endogenous p53, 120 min post-CHX treatment (Supplementary Fig. 10b). Data were analyzed using one-way ANOVA followed by Tukey's post hoc test and are presented as mean values ± SD; $n = 3$ (pWT, AcK89, AcK171), or 4 (GFP, AcK403) biologically independent samples.

**e, f** G6PD K403 acetylation-dependent interaction with p53' regulates Fyn-mediated G6PD phosphorylation at substoichiometric amounts of p53'. Representative Western blot (**e**) and quantification (**f**) of G6PD and p53' phosphorylation levels following in vitro incubation with active Fyn at indicated p53':G6PD molar ratios. Data were analyzed using one-way ANOVA followed by Tukey's post hoc test and are presented as mean values ± SD; $n = 6$ or 7 independent experiments (Supplementary Fig. 10e). **g** p53' impairs Fyn-dependent G6PD phosphorylation at stoichiometric p53':G6PD ratio. Representative Western blot showing p53' and G6PD phosphorylation levels, following incubation with active Fyn. **h** G6PD K403 acetylation results in increased Bax levels and decreased Bcl2 levels in HEK293T cells. **i** G6PD K403 acetylation correlates with increased cleaved caspase-3 levels in HEK293T cells. **j** Annexin V/DAPI double staining of HEK293T cells expressing pWT or AcK403. Data were analyzed using two-sided *t*-test and are presented as mean values ± SD; $n = 3$ biologically independent samples. ns not significant, ND not detected. Source data are provided as a Source Data file.

(Supplementary Fig. 11c), and cannot be attributed to differences in cellular NADPH levels (Supplementary Fig. 11d). Collectively, these data imply a pro-apoptotic effect of G6PD K403 acetylation. Importantly, flow cytometry analysis found an increase in annexin V-positive/DAPI-negative cells expressing AcK403, relative to pWT G6PD (Fig. 6j and Supplementary Fig. 11e, f). Taken together, we show that G6PD K403 acetylation results in the accumulation of Bax, and induction of pro-apoptotic signaling.

## Discussion

High throughput mass spectrometry studies have revealed that G6PD is acetylated at multiple sites. However, the effect and significance of individual acetylation sites are mostly unknown or partially inferred from Lys-to-Gln mutational analyses[34,35,38]. This gap in knowledge can be attributed partly to the technical challenges in acquiring biochemical evidence for individual acetylation events. Our research, using site-specifically acetylated G6PD, revealed that acetylation has an effect on G6PD's structure, function, and stability, as well as PTM and ability to mediate protein-protein interactions and activate non-metabolic processes. Specifically, we discovered that K89 acetylation activates G6PD and promotes its ubiquitylation, while K403 acetylation deforms the dimer interface and the structure of the distant active site. Importantly, we show that K403 acetylation leads to a p53-dependent increased cellular levels of Bax and induction of apoptotic signaling cascades. Together with current knowledge, we can now suggest a model for the effect of G6PD K89 and K403 acetylation on G6PD catalytic and cellular activities (Fig. 7).

Growing evidence shows that metabolism and cellular signaling are tightly connected, allowing modulation of cellular processes, such as cell proliferation and death, in response to the metabolic status of the cell. Acetylation is linked to the acetyl-CoA/CoA ratio, while sirtuin deacetylases depend on the NAD+/NADH ratio, suggesting that protein acetylation could serve as a metabolic rheostat. The PPP is important for the biosynthesis of nucleotides, amino acids, and fatty acids; thus, highly proliferating cells require increased metabolic flux through the PPP. In addition, as a major source of reducing power in the form of NADPH, the PPP can protect cells from apoptosis by preventing the oxidation of cytochrome c, which is important for caspase activation.

Therefore, inhibition of G6PD by, for example, K403 acetylation—and the subsequent downregulation of the PPP—can be expected upon induction of apoptosis. However, according to our data, the acetylation of G6PD served as a trigger for the induction of apoptosis and not vice versa. This is an intriguing example in which a single acetylation event of a metabolic enzyme can potentially downregulate a metabolic pathway and, at the same time, contribute to the activation of a signaling pathway, in this case, induction of apoptosis.

At first sight, a possible mechanism for the observed K403 acetylation-dependent increase in apoptotic signaling is the lack of G6PD activity that renders the cells more sensitive to oxidative damage. For example, G6PD-deficient mononuclear cells are more susceptible to redox imbalance-induced apoptosis[59]. However, the pro-apoptotic effect of K403 acetylation cannot be simply attributed to the inactivation of G6PD since it was observed in the absence of KDACi, which results in efficient deacetylation of AcK403 and reactivation of G6PD. Current knowledge shows that, unlike phosphorylation, acetylation is not amplified in cascades of acetylation events. Therefore, the stoichiometry of acetylation on a given lysine residue is expected to attenuate the magnitude of its effect. However, acetylation of non-histone proteins is considered a low-stoichiometry modification; specifically, in HeLa cells, the fraction of G6PD acetylated at position K403 was found to be 0.02%[52,60]. Nevertheless, although in the absence of KDACi AcK403 was deacetylated by endogenous KDACs, it resulted in increased Bax levels, decreased BCL2 levels, and induction of apoptotic signaling cascades. Hence, even at low-stoichiometry, acetylation can enable crosstalk between physiological processes in the cell, especially if the effect is irreversible and cumulative.

We found that K403 acetylation stabilizes the interaction with p53 and that the observed induction of apoptotic signaling is mediated by p53. The p53 transcription factor has diverse cellular functions, such as induction of apoptosis, senescence, and DNA repair. p53 also plays a role in metabolic regulation by several mechanisms, including transcriptional regulation and direct interaction with metabolic enzymes[56,61]. Specifically, the interaction between p53 and G6PD was suggested to render the latter inactive and monomeric[56]. In light of our data, the previously measured p53-dependent decrease in G6PD dimerization and activity may be explained by the fact that the isolated

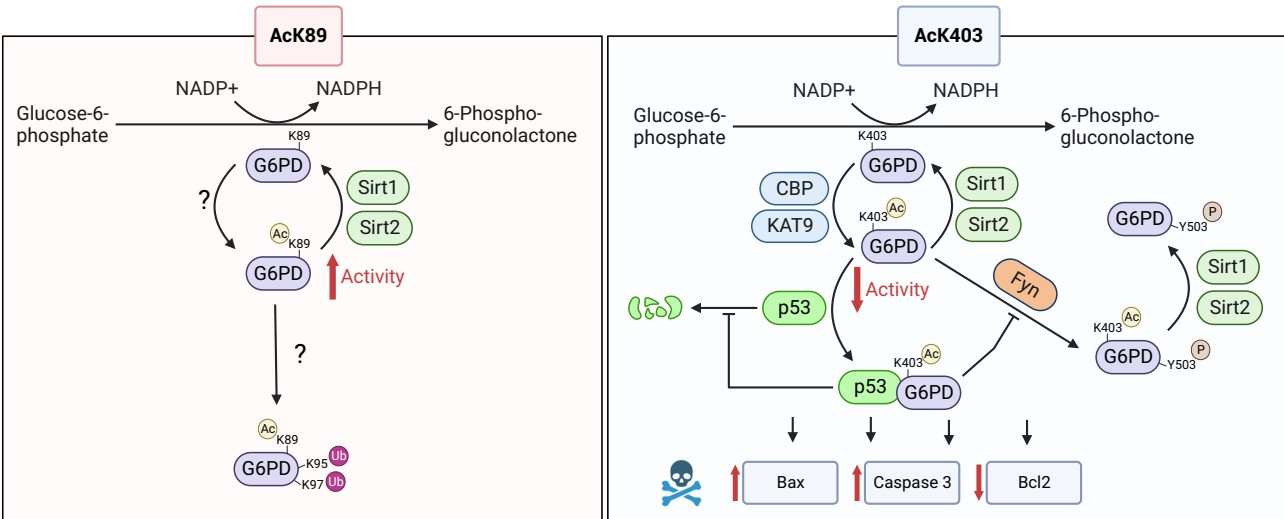

**Fig. 7 | Suggested model for the effect of K89 and K403 acetylation on activity and cellular functions of G6PD.** Left: Acetylation of K89 enhances the catalytic activity of G6PD and promotes its ubiquitylation on residues K95/K97. Sirt1 and Sirt2 can potentially deacetylate AcK89. Right: K403 is acetylated by KAT9 (Wang et al.[34]) and CBP (this work) and deacetylated by Sirt2 (Wang et al.[34] and this work) and Sirt1 (this work). Acetylation of K403 inhibits G6PD activity and promotes the interaction with p53, leading to the stabilization of p53 and induction of pro-apoptotic signaling. K403 acetylation also promotes Fyn-dependent phosphorylation of Y503, which can be partially inhibited by the interaction between K403-acetylated G6PD and p53. Y503 phosphorylation has no effect on K403 deacetylation by Sirt1 and Sirt2. Created with BioRender.com.

population of p53-bound G6PD was enriched by inactive, and mostly monomeric, K403-acetylated G6PD. Future research should characterize the interaction between AcK403 G6PD and p53 at the atomic level, to provide the molecular mechanism of this interaction. Moreover, regulation of p53 cellular level is mostly ascribed to ubiquitylation-dependent degradation, via MDM2 and other E3-ligases. A detailed understanding of the interaction between AcK403 G6PD and p53 may clarify if it directly stabilizes the latter by, for example, preventing its ubiquitylation.

Cross-talk between different PTMs can modulate the effects of individual modifications, and understanding the interplay between PTMs is therefore important for deciphering the combinatorial PTM pattern of proteins. Numerous phosphoproteomic studies found G6PD to be phosphorylated on residue Y503. However, the kinase responsible for Y503 phosphorylation, and the upstream signal leading to this event, are unknown. We found that G6PD Y503 is a substrate of Fyn kinase−known to phosphorylate Y401−and that K403 acetylation promotes Y503 phosphorylation by Fyn. A picture emerges in which the binding of p53 to G6PD sequesters Fyn. However, at low p53 levels, K403 acetylation favors G6PD binding to Fyn, resulting in a higher preference for G6PD phosphorylation over p53 phosphorylation. We can suggest a hypothetical mechanism for these observations, in which protein-protein interactions are determined by modular and PTM-dependent affinities between G6PD, p53, and Fyn, that may even involve the formation of a ternary complex. However, additional studies are required to fully understand the complex crosstalk between acetylation and phosphorylation and its effect on protein-protein interactions.

Residue K89 is not part of the active site or other functionally important elements, yet, it is highly conserved from *E. coli* to mammals. The molecular mechanism of K89 acetylation-dependent activation of G6PD is yet to be defined. Still, in Jurkat cells, the estimated intracellular concentration of G6P is 100 μM[62], while the measured $K_M^{app}$ of AcK89 for G6P was $124 \pm 4$ μM. Thus, K89 acetylation can enhance G6PD activity under physiologically relevant conditions. Moreover, 6-phosphogluconate dehydrogenase (6PGD), the second NADPH-producing enzyme in the PPP, is also activated by acetylation[63]. Hence, theoretically, coordinated acetylation of two key enzymes in the PPP (G6PD and 6PGD) can accelerate the metabolic flux through the PPP, in general, and the synthesis of NADPH in particular.

Functional studies of metabolic enzyme regulation by acetylation, especially in mammalian cells, usually rely on using Lys-to-Gln mutational analyses. Our study highlights the importance of studying acetylated proteins. For example, K89Q mutation had no effect on G6PD activity, although AcK89 G6PD displayed higher $V_{max}$. Similarly, the effect of K403 acetylation on G6PD structure and interaction with p53 could be identified only with site-specifically acetylated G6PD and not with K403Q G6PD. Moreover, direct in vivo and in vitro deacetylation measurements can be performed only with acetylated proteins. To enable measurements in mammalian cells, we introduced the concept of pWT as the non-acetylated control, which takes into account expression levels and unknown physiological effects of amber suppression.

In summary, through multiple lines of evidence, obtained both in vitro and in cultured cells, we demonstrate that G6PD acetylation is a multifaceted PTM that can control enzymatic activity and promote the induction of apoptotic events. This work emphasizes not only the importance of studying site-specifically acetylated proteins, but also the fact that single acetylation sites can coordinate between metabolic regulation and pivotal cellular processes such as apoptosis.

## Methods
### Reagents
Nε-acetyl-lysine (AcK) was purchased from Chem-Impex International (Wood Dale, IL). Diaphorase (#D5540), resazurin (#199303), NAD+ (#N7004), NADP+ (#N5755), G6P (#G7879), DSS (#S1885), CHX (#C7698), SAHA (#SML0061), NAM (#72340), MG132 (#M7449), phenylmethanesulfonyl fluoride (#P7626), sodium fluoride (#S1504), sodium orthovanadate (#S6508), ATP (#A2383), and 1 M manganese(II) chloride solution (#M1787) were obtained from Sigma-Aldrich. Sequencing grade chymotrypsin (#11418467001) and DNase I (#10104159001) were purchased from Roche Diagnostics (Mannheim, Germany). InstantBlue Coomassie Protein Stain (#ab119211) was purchased from Abcam (Cambridge, UK). Protein quantification was performed with QPRO-BCA Kit Standard from Cyanagen (Bologna, Italy). HisPur Ni-NTA Resin (#TS-88222) was purchased from Thermo Scientific (Waltham, MA). Alexa Fluor 647-conjugated Annexin V (#A23204) was purchased from Invitrogen (Waltham, MA). Aprotinin (#616370), Leupeptin (#108975), Pepstatin A (#516481), and NADPH (#481973) were obtained from EMD Millipore. DNA oligomers were obtained from Sigma-Aldrich unless otherwise stated. Enzymes and buffers for molecular biology were purchased from NEB (Ipswich, MA, USA).

### General
*E. coli* DH10B cells were used for plasmid amplification. Plasmid DNA was purified using spin columns from Macherey Nagel (Düren, Germany). All cloning steps were validated by DNA sequencing. When required, protease inhibitors were used at the following concentrations: 1.2 μg/mL leupeptin, 1 μM pepstatin A, 100 μM phenylmethanesulfonyl fluoride (PMSF), and 1 μg/mL aprotinin. Working concentrations of other inhibitors during cell lysis were as follows: deacetylase inhibitors (20 mM NAM, and 10 μM SAHA), phosphatase inhibitors (5 mM NaF and 1 mM Na₃VO₄). Details of all primers used in this work are listed in Supplementary Table 4.

### Plasmid construction
Robust transient expression of acetylated proteins in cultured mammalian cells was achieved using a pBud-based single vector carrying the genes of an orthogonal tRNA synthetase/amber suppressor tRNA pair for the cotranslational incorporation of AcK into TAG mutants of G6PD[39,43]. The full nucleotide sequence encoding AcKRS1-TAG-P2A-eGFP was cloned between BamHI and EcoRI restriction sites downstream of the CMV promoter. A tRNA cassette composed of a U25C mutant of the pyrrolysine tRNA_CUA downstream of a U6 promoter (IDT, Leuven, Belgium) was cloned into the same vector using the NheI restriction site[64]. An shRNA cDNA for continuous knockdown of endogenous G6PD (IDT; 5′-TCAGTCGGATACACACATATT-3′) was cloned into the pBud-based vector downstream of a U6 promoter. The DNA fragment encoding sh-resistant G6PD with a C-terminal 3 × Flag tag (IDT) was cloned into the same vector downstream of the EF1α promoter. An amber stop codon (TAG) mutation was introduced into the *g6pd* gene at the position of interest by overlapping PCR to give pBud-G6PD^Q83TAG, pBud-G6PD^K89TAG, pBud-G6PD^K95TAG, pBud-G6PD^K97TAG, pBud-G6PD^K171TAG, pBud-G6PD^K386TAG, pBud-G6PD^K403TAG, pBud-G6PD^K408TAG, pBud-G6PD^N414TAG, pBud-G6PD^K432TAG, and pBud-G6PD^K497TAG. Likewise, pBud-G6PD K89R, K89Q, K403R, K403Q, K89TAGK95R, K89TAGK97R, K89TAGK95/97R, N414TAGK95R, N414TAGK97R, N414TAG-K95/97R, Y401F, N414TAGY401F and K403TAGY401F were generated by overlapping PCR. Y503F, N414TAGY503F, and K403TAGY503F were introduced via one-step PCR mutagenesis.

For expression of G6PD in *E. coli*, the DNA fragment encoding G6PD WT (Addgene plasmid #41521)[56] with C-terminal 6×His tag was cloned into a pCDF vector, carrying a gene for the U25C mutant of pyrrolysine tRNA, downstream to lpp promoter[64]. In-frame TAG mutations at putative acetylation sites of G6PD were introduced by overlapping PCR to give pCDF-G6PD^N414TAG, pCDF-G6PD^K89TAG, pCDF-G6PD^K386TAG, pCDF-G6PD^K403TAG, pCDF-G6PD^K408TAG, pCDF-G6PD^K432TAG, and pCDF-G6PD^K497TAG. Likewise, Y401F and K403TAGY401F were generated by overlapping PCR. Y503F and K403TAGY503F mutations

were introduced via one-step PCR mutagenesis. For the expression of Fyn in *E. coli*, *fyn* and *fyndn* genes were amplified from plasmids pRK5-Fyn (Addgene plasmid #16032) and pRK5-DN Fyn (Addgene plasmid #16033), respectively[65]. For the cloning of *hdac6* and *p53'* genes, pBJ5-HDAC6 (kind gift from Dr. Zhang's lab, Karmanos Cancer Institute) and pCDNA3.1-p53 were used as templates, respectively[57,66].

## Cell lines and culture conditions

HEK293T (internal stock) and COS7 (internal stock) cells were cultured in complete DMEM (Biological Industries, Israel), and HCT116 (kind gift from Prof. Moshe Oren, Weizmann Institute, Israel) cells were cultured in McCoy's 5A medium (Biological Industries). Culture media were supplemented with 10% heat-inactivated fetal bovine serum (FBS), 2 mM L-glutamine, 1 mM sodium pyruvate, 100 U/mL penicillin G sodium, and 0.1 mg/mL streptomycin sulfate. Cultures were maintained at 37 °C in a humidified atmosphere containing 5% $CO_2$. All cell lines were confirmed to be mycoplasma-free using the EZ-PCR Mycoplasma Detection Kit (Biological Industries).

## Overexpression of G6PD in mammalian cells

Where indicated, mammalian cells were cultured in the presence of KDACi (3 μM SAHA, and/or 15 mM NAM). Stock solutions of KDACi (2 mM SAHA in DMSO, and 1 M NAM in DDW) were diluted in complete cell culture media immediately before usage, and media were filter sterilized (0.2 μm). As a control (minus KDACi), cell culture media were supplemented with the respective solvent only.

For the expression of site-specifically acetylated G6PD in mammalian cells, HEK293T or HCT116 cells (250,000) were seeded in each well of a 12-well plate. On the next day, and at least 30 min before transfection, media were replaced with complete DMEM or McCoy's 5A supplemented with 2 mM AcK. Transfection of HEK293T or HCT116 cells was carried out using LipoD293 (SignaGen Laboratories, Rockville, MD) or FuGENE HD (Promega, Madison, WI), respectively, according to the manufacturer's protocol. When cultures were incubated in the presence of KDACi, HEK293T cultures were supplemented with indicated inhibitors before transfection (together with AcK), while HCT116 cultures were supplemented with indicated inhibitors 10–12 h post-transfection. The transfected cells were then cultured in the presence of KDACi for another 48 h.

For enzymatic activity measurements in cell lysates, cells were collected and resuspended in lysis buffer composed of 50 mM Tris (pH = 7.4), 150 mM NaCl, 0.1% NP-40, protease inhibitors cocktail, deacetylase inhibitors, and 4 μg/mL DNAse I. The lysate was centrifuged at 21,000 × g at 4 °C for 10 min. The total protein concentration in the supernatants (total lysate) was determined by the BCA method. Levels of overexpressed G6PD were examined by Western blotting. Up to 1 μg of total lysate was used in enzymatic assays.

## Subcellular fractionation

HEK293T cells transfected with plasmids expressing acetylated G6PD variants were cultured in the presence of KDACi for two days. Total, cytosolic, and nuclear fractions were isolated following published protocols[67]. Briefly, cells were washed once with ice-cold phosphate-buffered saline (PBS) before harvest. Following one round of freeze-thaw cycle, cell pellets were resuspended in 420 μL of resuspension buffer (PBS containing 0.1% NP-40, and protease and deacetylase inhibitors) and incubated on ice for 20 min. Next, 140 μL of the total lysate was transferred into a new tube as the total fraction. The remaining lysed cell suspension was centrifuged at 10,000 × g for 30 s at 4 °C. 140 μL of the supernatant was saved as the cytosolic fraction. The pellet containing nuclei and residual cytoplasmic proteins was washed three times with resuspension buffer. The washed pellet was then resuspended in 70 μL of 2× reducing Laemmli sample buffer (Bioprep, #LSB4-R-50ml) and saved as the nuclear fraction. Nine μL of total and cytoplasmic fractions and 3 μL of nuclear fractions were separated by SDS-PADE and analyzed by Western blotting using indicated antibodies.

## Immunofluorescence

COS7 cells were plated at low density on #1.0 coverslips (Menzel, Braunschweig, Germany). The following day, cells were transfected with plasmids encoding the expression of indicated Flag-tagged G6PD variants in the presence of 10 mM NAM and 1 μM SAHA. 24 h later, cells were fixed using 4% paraformaldehyde, permeabilized with 0.5% Triton ×-100 for 10 min, and blocked with 10% FBS for 10 min. Cells were stained with a primary anti-Flag antibody (Sigma-Aldrich, #F1804, Clone M2; 1:500 dilution) for 3 h at room temperature, followed by a secondary Alexa Fluor 594 anti-mouse antibody (Thermo Fisher Scientific, #R37115; 1:1000 dilution). Finally, cells were mounted with DAPI Fluoromount-G (SouthernBiotech, Birmingham, AL). Immuno-labeled cells were imaged in 3D using a confocal spinning-disk microscope (Marianas; Intelligent Imaging, Denver, CO) with a 63× oil objective (numerical aperture, 1.4) and an electron-multiplying charge-coupled device camera (pixel size, 0.079 μm; Evolve, Photometrics, Tucson, AZ). Images were processed using SlideBook version 6 (Intelligent Imaging).

## Flow cytometry analysis of apoptotic cell death

HEK293T cells transfected with plasmids expressing acetylated G6PD variants were cultured for 48 h. Cells were harvested following standard trypsinization protocol, washed once with PBS and once with 1× binding buffer (10 mM HEPES pH = 7.4, 140 mM NaCl, 2.5 mM $CaCl_2$), and then stained with Alexa Fluor 647-conjugated Annexin V. After 10 min incubation in the dark, cells were pelleted and resuspended in 1× binding buffer containing 1 μg/mL DAPI. Flow cytometry data collection was performed using FACSAria II (BD Biosciences, Franklin Lakes, NJ) and CytExpert (version 2.5), followed by analysis using Kaluza Analysis (version 2.1). The total percentage of cells positive for Annexin V, but negative for DAPI, were considered early apoptotic cells.

## Bacterial expression and purification of recombinant proteins

**Purification of recombinant G6PD.** *E. coli* BL21(DE3) cells (NEB) were co-transformed with pBK plasmid for the expression of the evolved AcKRS3 synthetase[39], and pCDF-G6PD plasmid carrying a TAG mutation at the indicated site. Cells were cultured overnight at 37 °C in standard Luria-Bertani (LB) broth supplemented with 50 μg/mL spectinomycin and 50 μg/mL kanamycin. On the next day, the overnight culture was inoculated into 2 L of standard terrific broth (TB) medium supplemented with 50 μg/mL spectinomycin and 50 μg/mL kanamycin. At $OD_{600}$ = 0.4, 10 mM AcK and 20 mM NAM were added. Protein expression was induced with the addition of 0.5 mM isopropyl $\beta$-D-thiogalactopyranoside (IPTG) at $OD_{600}$ = 0.7. After 4 h incubation at 37 °C, the cells were harvested by centrifugation (4750 × g) and resuspended in Ni-NTA buffer [25 mM HEPES pH = 8, 150 mM NaCl, 15 mM $\beta$-mercaptoethanol ($\beta$-ME), 20 mM imidazole] supplemented with 1 mM PMSF and 20 mM NAM. Cells were lysed by sonication, and the lysate was centrifuged at 10,000 × g for 1 h at 4 °C. The supernatant was loaded onto a 5 mL HisTrap HP column (Cytiva, Marlborough, MA) pre-equilibrated with Ni-NTA buffer. G6PD was eluted using a linear gradient (0–100%) of Ni-NTA elution buffer (25 mM HEPES pH = 8, 150 mM NaCl, 15 mM $\beta$-ME, 500 mM imidazole). Fractions containing G6PD were identified by SDS-PAGE, and concentrated using a Centricon with a 10 kDa cutoff membrane (Merck Millipore, Burlington, MA). The concentrated sample was then purified by size exclusion chromatography using a HiLoad 26/600 Superdex 200 column (Cytiva), pre-equilibrated with SEC buffer [10 mM Tris pH = 7.5, 150 mM NaCl, 5 mM dithiothreitol (DTT), 10% Glycerol]. Fractions containing G6PD were pooled and concentrated. The final concentration of G6PD was determined by the Bradford method, and the

protein was stored in 30% glycerol at -80 °C. The purity and identity of the recombinant enzymes were verified by SDS-PAGE followed by coomassie staining and Western blotting, respectively.

**Purification of recombinant p53.** Quadruplet mutant of p53' was expressed as a fusion protein with N-terminal 6×His tag, followed by the lipoyl domain and tobacco etch virus (TEV) protease cleavage site, and purified using previously published protocols[42]. Briefly, p53' was expressed in *E. coli* BL21 strain induced with 1 mM IPTG, and incubated overnight at 22 °C. Cells were lysed in 50 mM phosphate buffer pH = 8, 300 mM NaCl, 10 mM imidazole, and 10 mM $\beta$-ME, and the clear lysate was loaded onto a 1 mL HisTrap HP column (Cytiva). The fusion protein was eluted with a 50 mM phosphate buffer pH = 8, containing 300 mM NaCl, 10 mM $\beta$-ME, and 250 mM imidazole. TEV protease was added to the eluted fractions, and the sample was dialyzed overnight at 4 °C with gentle stirring against dialysis buffer (25 mM Tris buffer pH = 7.5, 300 mM NaCl, 10% glycerol, 10 mM $\beta$-ME). The dialyzed protein solution was diluted 1:10 in cold heparin buffer A (25 mM potassium phosphate pH = 7.5, 10 mM $\beta$-ME, 10% Glycerol) and loaded onto a heparin column (Cytiva). p53' was eluted with a linear gradient over 20 column volumes of heparin buffer B (25 mM potassium phosphate pH = 7.5, 10 mM $\beta$-ME, 10% Glycerol, 1 M NaCl). Fractions containing p53' were further purified by size exclusion chromatography (25 mM phosphate buffer pH = 7.2, 300 mM NaCl, 5 mM DTT, and 10% glycerol).

**G6PD activity and kinetic characterization**
Enzymatic activity of G6PD was determined using a diaphorase-resazurin coupled assay as previously described[47]. The reaction mixture was composed of 50 mM Tris (pH = 7.4), 3.3 mM MgCl$_2$, 1 U/mL diaphorase, 0.1 mM resazurin, and up to 50 µM NADP$^+$ and 1600 µM G6P, in a total volume of 100 µL. Reactions were initiated by adding cell lysates or recombinant protein (final concentration of 1.25 nM). Enzyme activities were measured by monitoring the increase in fluorescence (Ex. 530 nm, Em. 580 nm) using a microplate reader (Tecan, Männedorf, Switzerland). Initial velocity data were obtained by varying the concentration of NADP$^+$ (0–50 µM) with a constant G6P concentration (1600 µM), or by varying the concentration of G6P (0–1600 µM) with a constant NADP$^+$ concentration (50 µM). The apparent $K_M$ ($K_M^{app}$) and maximum velocity ($V_{max}^{app}$) values were calculated by fitting the initial velocity data to the Michaelis-Menten equation using R. The value of $k_{cat}^{app}$ was calculated from $V_{max}^{app}$ using a molecular weight of 59 KDa for the G6PD monomer.

The following measures were taken to ensure enzyme stability throughout the experiment. The stock solution of recombinant G6PD was first diluted to 16.7 nM with G6PD dilution buffer (50 mM Tris pH = 7.4, 150 mM NaCl, and 0.1% NP-40). However, we found that the diluted enzyme lost about 70% of activity within 1 h at 4 °C. In contrast, when WT G6PD was diluted to 16.7 nM in G6PD dilution buffer supplemented with 1 µM of NADP$^+$, the enzyme remained fully active for at least 3 h. Therefore, to achieve reliable, linear initial-rate traces for detailed kinetic analyses, concentrated G6PD was first diluted to 16.7 nM in G6PD dilution buffer supplemented with 1 µM NADP$^+$. To initiate the enzymatic reaction, 7.5 µL of 16.7 nM G6PD in dilution buffer was mixed with 92.5 µL of the activity reaction mixture, thereby creating an enzyme solution at a final concentration of 1.25 nM. The addition of NADP$^+$ to G6PD dilution buffer changed the final NADP$^+$ concentration by only 0.075 µM, which was included when calculating final NADP$^+$ concentrations.

**In vitro cross-linking**
30 nM of purified G6PD variants in PBS, supplemented with 5 mM DSS and 1 µM NADP$^+$, were incubated at room temperature for 30 min. The samples were boiled in SDS sample buffer and subjected to immunoblotting with anti 6×His antibody. Levels of monomeric and dimeric

forms of G6PD were determined from immunoblot intensities, and the relative level of dimerization was determined from the ratio between the dimeric form and total protein (i.e., the sum of monomeric and dimeric forms).

**Thermostability assay**
12.5 nM of recombinant enzymes in PBS, supplemented with 1 µM or 50 µM NADP$^+$, was incubated at indicated temperatures (between 25 and 70 °C) for 20 min and transferred to an ice bath for 5 min. Then, 10 µL of the sample was added to 90 µL of activity assay reaction buffer composed of 50 mM Tris (pH = 7.4), 3.3 mM MgCl$_2$, 1 U/mL diaphorase, 0.1 mM resazurin, 50 µM NADP$^+$, and 1600 µM G6P (1.25 nM of recombinant G6PD). Enzymatic activity was measured in 100 µL reactions as described above. Residual activity was plotted relative to the activity measured after incubation at 25 °C, and $T_{1/2}$ value−the temperature at which the enzyme retains half of its original activity−was calculated by fitting the data to a four-parameter logistic function using R.

**Aggregation assay**
Aggregation of purified G6PD variants was measured by following the absorbance at 340 nm over time. Measurements were performed using a covered flat-bottom 96-well plate (Corning), with 100 µL of 3 mg/mL G6PD variants in SEC buffer (10 mM Tris pH = 7.5, 150 mM NaCl, 5 mM DTT, and 10% Glycerol) supplemented with indicated NADP$^+$ concentration. Plates were placed in a Spark multimode microplate reader, and absorbance at 340 nm was measured every 15 min (following 30 s mixing). All experiments were performed at a constant temperature of 37 °C.

**Chymotrypsin digestion**
200 ng of recombinant G6PD were incubated at room temperature with 50 ng of chymotrypsin in 200 µL of chymotrypsin digestion buffer (100 mM Tris pH = 8, 10 mM CaCl$_2$). Samples were taken at indicated times, and reactions were quenched by boiling in 1× reducing Laemmli sample buffer. Similarly, samples of G6PD were incubated for 1 h at room temperature with increasing amounts of chymotrypsin (50 ng, 100 ng, 250 ng, and 500 ng). The levels of remaining full-length G6PD, as a function of time or chymotrypsin concentration, were examined by Western blotting.

**CHX chase assay**
50,000 HCT116 cells per well were seeded in each well of a 48-well plate and cultured overnight. The next day, cells were transiently transfected with plasmids encoding the expression of site-specifically acetylated G6PD. Twelve hours post-transfection, the cells were treated with 100 µg/mL of CHX and cultured for the indicated time (0–36 h). Cells were then collected and lysed in RIPA buffer (50 mM Tris buffer pH = 8, 150 mM NaCl, 1% v/v Triton X-100, 0.5% w/v sodium deoxycholate, and 0.1% w/v SDS) containing DNAse I and protease inhibitors. The total cell lysate was clarified by centrifugation (21,000 × g, 4 °C, 10 min), and 10 µg of total protein was resolved by SDS-PAGE. G6PD levels were examined by Western blotting and normalized to enolase content.

The stability of endogenous p53 was monitored by transfecting HCT116 cells with acetylated G6PD expression plasmid (pWT, AcK89, AcK171, AcK403) or control vector (GFP). Culture media was exchanged 12 h post-transfection, and 46 h post-transfection cells were treated with 50 µg/mL CHX for the indicated time, collected, and subjected to Western blot analysis. p53 level was normalized to the total protein load using in-gel 2,2,2-trichloroethanol (TCE) fluorescence (as detailed below for the general immunoblotting procedure).

**Immunoprecipitation**
Total cell lysates containing about 300 µg of protein were used for Flag immunoprecipitation. Anti-Flag M2 magnetic beads (Sigma-Aldrich,

#M8823) were added to the supernatant, and samples were incubated overnight at 4 °C with gentle mixing. Beads were washed three times with lysis buffer, and bound proteins were eluted using 1× non-reducing Laemmli sample buffer (Bioprep, #LSB4-NR-5ml) at 70 °C for 10 min and analyzed by immunoblotting.

## Immunoblotting

Protein lysates and eluates were resolved on 12% SDS-PAGE gels supplemented with 1% TCE. Following electrophoresis, proteins were visualized by illuminating the gel with UV light to quantify the total protein load. Proteins were then transferred to a nitrocellulose membrane (Cytiva) using a semi-dry transfer apparatus (Powerblotter station, Invitrogen). The membrane was blocked for 1 h at room temperature with 5% non-fat dry milk in TBST buffer (Bio-Lab, Israel) before overnight incubation with the indicated primary antibody at 4 °C. Following washing with TBST, the membrane was incubated with an HRP-conjugated secondary antibody for 1 h at room temperature. Proteins were visualized with ECL Reagent (Cytiva, #RPN2106) using the Fusion FX imaging system (Vilber, France). Uncropped blots are provided in the Source Data file.

The following primary antibodies were used for Western blotting: anti-His Tag (Applied Biological Materials, #G020; 1:4000), anti-HA Tag (Applied Biological Materials, #G166; 1:1000), anti-G6PD (Abcam, #ab993; 1:5000), anti-p53 (Abcam, #ab90363, Clone pAb122; 1:1000), anti-p53 (Abcam, #ab1101, Clone DO-1; 1:1000), anti-acetyl lysine (Abcam, #ab21623; 1:2500), anti-acetyl lysine (Cell Signaling Technology, #9681; 1:1000), anti-phospho-tyrosine (Cell Signaling Technology, #96215; 1:1000), anti-Fyn (Cell Signaling Technology, #4023; 1:1000), anti-Flag (Sigma-Aldrich, #F1804; 1:1000), anti-Ubiquitin (Cell Signaling Technology, #3936; 1:1000), anti-ENO1 (Abcam, #ab155102; 1:20000), anti-Histone 3 (Abcam, #ab1791; 1:4000), anti-alpha-tubulin (Millipore, #CP06; 1:1000), anti-Bax (Abcam, #ab32503; 1:500), anti-Cleaved Caspase-3 (Cell Signaling Technology, #9664; 1:1000), anti-Bcl-2 (Abcam, #ab196495; 1:1000). Unless stated otherwise, the detection of acetylated lysine residues by Western blotting was performed using the pan-anti-AcK antibody from Cell Signaling (#9681).

The following secondary antibodies have been used for Western blotting: Goat Anti-Mouse IgG H&L (HRP) preadsorbed (Abcam, #ab7068; 1:10000), Recombinant Protein G (HRP) (Abcam, #ab7460; 1:5000), Goat Anti-Rabbit IgG H&L (HRP) (Abcam, #ab6721; 1:5000).

## In vitro deacetylation assay

HEK293T cells were transfected with plasmids encoding the expression of Flag-tagged HDAC6, Sirt1, or Sirt2 and cultured for 48 h. Cells were harvested, resuspended in lysis buffer (50 mM Tris-HCl pH = 7.5, 0.5 M KCl, 1% NP-40, and 0.5 mM DTT) supplemented with protease and phosphatase inhibitors, and incubated on ice for 10 min. Following centrifugation at $21,000 \times g$ for 10 min, clear supernatants were incubated overnight with anti-Flag M2 beads at 4 °C. Beads were washed three times with lysis buffer and once with elution buffer (50 mM Tris-HCl buffer pH = 8, 4 mM MgCl$_2$, 50 mM NaCl, and 0.5 mM DTT). Proteins were eluted with 100 μL of elution buffer supplemented with 200 μg/mL 3× Flag-peptide (Sigma-Aldrich, #F4799) and stored at −80 °C until usage. In vitro deacetylation assay was performed in 300 μL of HDAC assay buffer (25 mM Tris pH = 8, 137 mM NaCl, 2.7 mM KCl, 1.0 mM MgCl$_2$, and 0.1 mg/mL BSA), using 12.5 μg of acetylated substrates and immunopurified HDAC6, Sirt1, or Sirt2, for either 1 h at 37 °C, or 4 h at 25 °C. For deacetylation assay with Sirt1 or Sirt2, the assay buffer was supplemented with 1 mM NAD$^+$. Wild-type G6PD was used as a control. Samples were taken at indicated time points, and the reaction was quenched by boiling in 1× reducing Laemmli sample buffer. The extent of deacetylation was quantified from the remaining acetylation level by immunoblotting using anti-AcK antibody. When required for G6PD activity measurements, the deacetylation reaction

was quenched by adding 20 mM NAM, and 15 ng of G6PD was immediately subjected to G6PD activity assay.

## In cell ubiquitylation assay

HEK293T cells were co-transfected with a plasmid encoding the expression of ubiquitin and indicated plasmid. Thirty-two hours post-transfection, cells were supplemented with MG132 (5 μM) or DMSO, and cultured for an additional 16 h. Proteins were immunoprecipitated from total cell lysates using anti-Flag beads and immunoblotted with the indicated antibodies.

## Pull-down assay

Twenty μg of recombinant His-tagged G6PD were mixed with 200 μg of total protein from HEK293T cell lysates overexpressing HA-tagged p53 in 400 μL of pull-down binding buffer (20 mM HEPES pH = 7.8, 50 mM KCl, 10% glycerol, 2.5 mM MgCl$_2$, 0.05% NP-40, 20 mM NAM and protease inhibitor mixture). The above mixture was divided into two fractions for HA and Ni-NTA pull-drown assays. In pull-down assays using anti-HA antibody, the mixture was incubated with anti-HA magnetic beads (Cell Signaling Technology, #11846S) for 1 h at 4 °C. Beads were washed twice with pull-down binding buffer, three times with ice-cold PBS, and boiled in 2× non-reducing Laemmli sample buffer. Protein samples were separated by 12% SDS-PAGE and analyzed by Western blotting. To avoid high chain IgG signals, immunoblotting using anti 6×His antibody was performed using protein G-HRP (Abcam, #ab7460; 1:5000 dilution). In pull-down assays using Ni-NTA, the binding buffer was supplemented with imidazole (10 mM) before incubation with Ni-NTA agarose beads (30 μL). The samples were incubated for 1 h at 4 °C with gentle mixing. The beads were precipitated by centrifugation ($2000 \times g$ for 3 min), washed five times with pull-down binding buffer containing 60 mM imidazole, and bound proteins were eluted by boiling in 2× reducing Laemmli sample buffer. Eluted proteins were analyzed by Western blotting using indicated antibodies.

## In vitro kinase assay

Bacterially expressed and purified His-tagged G6PD (460 ng) was incubated with 10 ng of activated GST-Fyn (Abcam, #ab84696) in kinase buffer (40 mM Tris pH = 7.4, 20 mM MgCl$_2$, 25 mM MnCl$_2$, 1 mM DTT, 0.1 mg/mL BSA, 10% glycerol, 1 μM NADP$^+$). The reaction was initiated by the addition of ATP at a final concentration of 50 μM. Reactions were performed in a total volume of 20 μL at room temperature for 40 min, and terminated with the addition of EDTA (final concentration of 50 mM), followed by addition of Laemmli sample buffer. When the assay was performed in the presence of p53', G6PD and p53' were first combined together at p53':G6PD molar ratio of 1:16, 1:8, 1:4, 1:2, and 1:1, before the addition of Fyn kinase to the reaction mixture. Tyrosine phosphorylation of substrates was evaluated by Western blotting using a pan-phosphotyrosine antibody.

## NADPH measurement

Intracellular levels of NADPH were measured by an enzyme-coupled assay, using recombinant G6PD and diaphorase in a cyclic NADP$^+$/NADPH reaction buffer containing resazurin. The assay was established based on a previously published protocol[68]. Briefly, HEK293T cells were cultured in 6-well plates and transfected with the indicated plasmids. Forty-eight hours post-transfection, cells were lysed in 150 μL extraction buffer (20 mM NAM, 20 mM NaHCO$_3$, 100 mM Na$_2$CO$_3$) supplemented with DNAse I and protease inhibitors. Cell lysates were cleared by centrifugation, and the total protein concentration in the supernatant fraction was determined by the BCA method. 100 μL of the supernatant were incubated at 60 °C for 30 min to inactivate G6PD (exo- and endogeneous) and decompose NADP$^+$, without affecting NADPH. Next, inactivated samples (60 μg of total protein in a final volume of 40 μL) were mixed with 60 μL of cyclic

NADP$^+$/NADPH reaction buffer (80 mM Tris-HCl pH = 7.6, 2.64 mM MgCl$_2$, 0.16 mM resazurin, 0.8 U diaphorase, and 1 nM of recombinant human G6PD) in a dark 96-well plate (Greiner Bio-one). After 1 min incubation in the dark at 28 °C, 20 μL of 10 mM G6P was added to the mixture, and the fluorescence of resorufin (Ex. 530 nm, Em. 580 nm) was monitored using a microplate reader. Samples were measured in triplicates, and the rate of resorufin production was considered proportional to NADPH concentration in the sample.

## X-ray crystallography
Crystallization of AcK89 and AcK403 G6PD was performed using the sitting drop vapor diffusion method by mixing 0.3 μL of protein solution (5.3 mg/mL AcK89 or 3 mg/mL AcK403) with 0.3 μL of crystallization solution and incubation at 20 °C. AcK89 G6PD was crystallized in a crystallization solution containing 0.2 M L-Proline, 0.1M HEPES pH = 7.2, and 20% PEG 3350. AcK403 G6PD was crystallized in a crystallization solution containing 5% Tacsimate, 0.1 M HEPES pH = 7.0, and 10% PEG MME 5000. Crystals were usually formed within two weeks. Crystals were flash-frozen in liquid nitrogen, and X-ray diffraction datasets were collected at the European Synchrotron Radiation Facility (Grenoble, France), beamline ID-29. Data were indexed and integrated with XDS using a previously solved G6PD structure (PDB: 6E08) as the search model[47]. Refinement and model building were performed using CCP4 Phoenix[69]. Figures were prepared using PyMol.

## LC-MS/MS
Flag-tagged AcK403 or pWT were co-expressed with Fyn or FynDN in HEK293T for 48 h in the presence of deacetylase inhibitors. Harvested cells were disrupted in lysis buffer (0.5 M KCl, 50 mM Tris-HCl pH = 7.5, 1% NP-40, and 0.5 mM DTT) containing protease and phosphatase inhibitors. Proteins were isolated using anti-Flag M2 beads as described above, and bound proteins were eluted using 2× non-reducing Laemmli sample buffer. Eluted proteins were separated by 12% Trisglycine SDS-PAGE, stained with 0.1% (w/v) Coomassie brilliant blue R250, and the band corresponding to G6PD was excised. The sample was reduced with 3 mM DTT (60 °C for 30 min), modified with 10 mM iodoacetamide in 100 mM ammonium bicarbonate (in the dark, room temperature for 30 min), and digested in 10% acetonitrile and 10 mM ammonium bicarbonate with modified trypsin (Promega) at a 1:10 enzyme-to-substrate ratio (overnight at 37 °C). The resulting peptides were desalted using C18 tips and subjected to LC-MS/MS analysis. The peptides were resolved by reverse-phase chromatography on 0.075 × 300-mm fused silica capillaries (J&W) packed with Reprosil reversed-phase material (Dr. Maisch GmbH, Germany). The peptides were eluted with a linear 30-minute gradient of 5% to 28% acetonitrile with 0.1% formic acid in water, 15 min gradient of 28% to 95% acetonitrile with 0.1% formic acid in water, and 15 min at 95% acetonitrile with 0.1% formic acid in water at a flow rate of 0.15 μL/min. Mass spectrometry was performed by Q Exactive plus mass spectrometer (Thermo) in a positive mode using a repetitively full MS scan followed by high collision dissociation (HCD) of the ten most dominant ions selected from the first MS scan. The mass spectrometry data was analyzed using Proteome Discoverer 2.4 software with Sequest (Thermo) algorithm against the Human UniProt database and Flag-tagged WT or pWT G6PD, with a mass tolerance of 10 ppm for the precursor masses and 0.05 amu for the fragment ions. Oxidation (M), phosphorylation (S, T, Y), acetylation (K), ubiquitination (K), methylation (K), and butyrylation (K) were accepted as variable modifications, and carbamidomethyl on Cys was accepted as a static modification. The minimal peptide length was set to six amino acids, and a maximum of two miscleavages was allowed. Peptide- and protein-level false discovery rates (FDRs) were filtered to 1% using the target-decoy strategy. Semi-quantitation was done by calculating the peak area of each peptide based on its extracted ion currents (XICs), and the area of the protein is the average of the three most intense peptides from each protein. Analysis was performed by the Smoler Proteomics Center at the Technion, Israel.

## Bioinformatic analyses
Evolutionarily conserved lysine residues were identified by multiple sequence alignment of G6PD protein sequences from selected organisms (P11413:*H. sapiens*, Q00612:*M. musculus*, P05370:*R. norvegicus*, P11412:*S. cerevisiae*, P12646:*D. melanogaster*, A0A2S1ZD90:*H. armigera*, A0A066R341:*E. coli*, A0A0J4W7K0:*K. quasipneumoniae*). Structural conservation (position of specific lysine residues) was verified manually by superimposing the relevant three-dimensional structures. For organisms with an unknown crystal structure of G6PD, the three-dimensional structure was predicted by AlphaFold[70]. Putative acetylation sites were identified based on previously published proteomic studies[18–32]. The effect of amino acid substitutions on protein function was predicted using the Sorting Intolerant from Tolerant tool (SIFT, http://sift.jcvi.org/)[71]. The conservation of the variant across species was assessed by Combined Annotation-dependent Depletion scores (CADD, http://cadd.gs.washington.edu/score)[72].

## Statistics and reproducibility
Statistical significance was calculated with R. Multiple comparisons were performed using a one-way ANOVA test with post hoc Tukey test, unless stated otherwise. Values of $p < 0.05$ were considered statistically significant. Representative results presented in Figs. 3a, c, 4b, f, and 6c were reproduced by two independent experiments. Representative results presented in Fig. 6g and h were reproduced by four and three independent experiments, respectively. Cellular localization of acetylated G6PD by confocal microscopy was based on at least ten random images of cells expressing a given G6PD variant. Levels of BAX, BCL-2, and cleaved caspase-3 were determined in two cell lines (HEK293T, HCT116) by at least two independent measurements. One sample of each indicated protein was analyzed by LC-MS/MS.

## Reporting summary
Further information on research design is available in the Nature Portfolio Reporting Summary linked to this article.

## Data availability
Atomic coordinates have been deposited in the RCSB Protein Data Bank [https://www.rcsb.org] under accession codes PDB ID 7ZVD (AcK89 G6PD) and PDB ID 7ZVE (AcK403 G6PD). The protein structures used for analysis in this study are available in the Protein Data Bank under accession codes 2BHL, 6E08, 6VA7, 7SEI. LC-MS/MS data have been deposited in the ProteomeXchange Consortium via the PRIDE partner repository with the dataset identifier PXD041775 (phosphorylated G6PD), and the dataset identifier PXD044906 (recombinant and immunopurified G6PD). Data supporting the findings of this study are available in the Supplementary Information file. Source data for Figs. 1–4, 6, and Supplementary Figs. 1–6, 9–11 are provided with this paper. Additional data can be provided by the corresponding author upon request.

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

## Acknowledgements

This work was supported by the European Research Council (ERC) under the European Union Horizon 2020 research and innovation program [grant number 678461 to E.A.]; and the Israel Science Foundation [grant number 1769/20 to E.A.].

## Author contributions

Conceptualization: F.W., and E.A.; investigation: F.W. performed most of the experiments, N.H.M. assisted with cloning, protein purification, and preliminary studies, I.D. assisted with confocal microscopy, O.K. assisted with flow cytometry measurements, A.S. assisted with protein crystallization; methodology: F.W.; formal analysis: F.W. and E.A., A.S. analyzed x-ray diffraction data; visualization: F.W.; writing–original draft: F.W., and E.A.; writing–reviewing and editing: F.W. and E.A., all authors read and approved the manuscript; supervision: N.E., R.G., and E.A.; funding acquisition: E.A.

## Competing interests

The authors declare no competing interests.
