## [Peer Review File · Nature Communications]

REVIEWER COMMENTS

Reviewer #1 (Remarks to the Author):

The manuscript by Wu et al, presents a very thorough study of lysine acetylation of glucose-6-phosphate dehydrogenase and its role on the regulation of enzymatic activity, and apoptotic signaling. Their conclusions are adequately supported by enzyme kinetics, immunochemistry, immunofluorescence, cellular, X-ray crystallography, and proteomics data which is technically sound. It was a great pleasure to read such methodologically precise and well written paper, I do not have any comments, and I am happy to recommend its publication.

Reviewer #2 (Remarks to the Author):

In the article "Acetylation-dependent coupling between G6PD activity and apoptotic signaling", Wu and co-workers investigate the molecular details of how acetylation at specific lysine residues modulates the activity of glucose-6-phosphate dehydrogenase (G6PD). Previous proteomics work has identified many acetylated lysines (AcKs) on G6PD under various condition. Most biochemical follow-up research has relied on site-directed mutagenesis to mimic acetylated (glutamine) or deacetylated (arginine) lysine. As Wu et. al. note, however, glutamine can sometimes be a poor mimetic for AcK. After prioritizing a handful of lysines based on multiple criteria, the authors use the amber codon method to incorporate AcK into G6PD at 100% stoichiometry onto specific sites. By performing both cell-based and in vitro experiments, they demonstrate opposing effects of AcK on two different sites in inhibiting (K403) and activating (K89) the enzyme. Furthermore, the authors show that the two contrasting acetyl marks participate in cross-talk with ubiquitination and phosphorylation. This work does a nice job of carefully characterizing complex PTM-mediated regulation of G6PD, which is quite complicated do to the PTM crosstalk. As this is an important enzyme for cancer biology and other research fields, I believe that the data will be of wide interest. My only primary concern is that the authors conclusions from acetyl immunoblots early in the paper assume that there is no crosstalk among acetyl sites, even though their model at the end of the paper suggests that G6PD is subject to extensive PTM crosstalk. A few relatively straight-forward mass spectrometry analyses would shore up this issue. Therefore, I recommend that the manuscript be reevaluated for publication in Nature Communications after the authors address a small number of my concerns and those of the other reviewers.

Primary Concerns:

1) Page 5 Line 22: How can the authors know for certain that all of the detected acetylation is indeed from the non-natural amino acid incorporation in each case? Is it not possible that some of the signal for specific AcK variants in Figure 1C is from endogenous acetylation on other lysines than the one containing the non-natural AcK? The authors show later in the paper that PTM crosstalk exists between AcK and ubiquitination and phosphorylation—couldn't it also exist between AcK on different sites by acetylation of one residue recruiting/repelling an acetyltransferase or deacetylase? I suggest the authors conduct unbiased LC-MS/MS analysis of at least a subset of these acetylated proteins treated with and without KDACi to support this assumption that the signal detected by the western blots is strictly from the single AcK sites.

Minor Concerns:

2) Page 2, Line 6: The authors may want to revise this sentence to read "Originally thought to be exclusively involved....acetylation is now additionally associated....", to make it clear that the idea that acetylation regulates histones has not changed, but acetylation is now known to also have a broader role in the cell.

3) Page 3, Line 15: Non-canonical amino acid incorporation is technically not dependent on there being evidence that the a PTM exists in vivo, but the rationale for developing hypotheses to test using previous data makes sense. Therefore, I suggest changing "known acetylation sites" to "any lysine, most commonly those already known to be acetylated from previous data."

4) Page, 4, Line 12: It was unclear to me how the pWT variants serve as appropriate controls, as "AcK was incorporated into a permissive site". I'm sure that I am missing something, but the authors should clarify by adding an additional sentence explaining how this works. It seems important that the pWT does not have any AcK incorporated as I see no signal on the western blot in Figure 1C.

5) Page 15, Line 48: While the authors did not identify a peptide including Y401 and K403, was this only tried with trypsin? This experiment could be repeated using another protease that may produce a more easily detectable peptide spanning these residues.

6) The authors are quite thorough in referencing appropriate work. Several missing papers that the authors could consider citing include a review that describes regulation of G6PD by PTMs (PMID: 36091812) and several primary articles on G6PD regulation by acetylation (PMID: 24769394; PMID: 27356773; PMID: 34310804).

7) Figure 7 legend: This figure is a nice visual summary of the study. However, the figure legend should be made to briefly describe all aspects of what the figure shows. This figure is very important for the reader to understand the take home from a lot of data.

Reviewer #3 (Remarks to the Author):

The manuscript by Wu et al., entitled "Acetylation-dependent coupling between G6PD activity and apoptotic signaling" reports the link between lysine acetylation-dependent G6PD activity and tumor cell apoptotic signaling. The authors first sought to determine the effect of lysine acetylation on G6PD activity using deacetylase inhibitors and determined that sirtuin-mediated lysine acetylation modulates G6PD activity. The authors then investigated the role of each of the 9 known acetylated lysines in G6PD using the point/site-mutated G6PD variants in G6PD knocked-down tumor cell lines. K403 was found to reduce, whereas K89 was found to increase G6PD enzymatic activity, and the balance of K89, K171, K386, K403, K408, and K432 modulates G6PD activity. Using *E. coli* expressed and purified G6PD variants, the author concluded that G6PD can be activated by a single lysine acetylation but is deactivated by multiple lysine acetylation. The author then studied the stability of the recombinant G6PD mutants and aggregation. The authors focused on K89 and K403, particularly K403 for its regulation by KATs and KDACs and its effects on G6PD conformations and phosphorylation, and determined that K403Ac deforms the dimer interface and renders G6PD monomeric. Further studies determined that K403Ac promotes p53 interaction and G6PD to stabilize p53. Finally, the authors observed that G6PD K403Ac promotes Bax accumulation through a p53-dependent mechanism, which leads to the conclusion that G6PD K403 acetylation promotes the induction of apoptotic signaling. Overall, the authors performed extensive biochemical and structural studies to determine lysine acetylation and G6PD function. PTM, including lysine acetylation, plays an important role in G6PD function, the finding that K403 of G6PD may be important for G6PD function in tumor cell apoptosis is interesting.

Major concerns

The major concern is the lack of new findings. All the studied lysine acetylation sites in G6PD are known in the literature. K403 acetylation in G6PD has been extensively studied as a negative regulator of G6PD activity. P53 also has been shown to bind to G6PD to prevent G6PD dimerization. The authors state that they found that K403 acetylation promotes the interaction between p53 and G6PD and stabilizes p53 in cells. However, what is new here in terms of G6PD dimer formation and function is not clear.

A major conclusion of this manuscript is the G6PD-p53-Bax-apoptotic signaling. The authors observed an increase in Bax in AcK403 G6PD-expressed cells, but not in pWT G6PD- cells. In parallel, HCT116 p53 WT cells have higher p53 than HCT116 p53 KO cells. These observations are correlative and no data to support the G6PD-p53-Bax-apoptosis pathway. Recent literature shows that there are compensatory pathways (ME1 and DH1) for G6PD and G6PD is not essential for certain tumor cell tumor cell proliferation and survival. A definitive experiment is to knock out G6PD to see if it is essential for tumor cell apoptosis here. In addition, G6PD/NADPH regulates HDAC3, a known apoptosis regulator. These may all affect the correlation observed in this study.

The author stated that: "Thus, using site-specifically acetylated G6PD, we show for the first time that G6PD can be activated and not only deactivated by the acetylation of a single lysine residue". The term "for the first time" should be used with caution. There are extensive literatures reporting lysine acetylation and G6PD activity. Only a small part of related references is cited in this manuscript. In addition, E coli-expressed G6PD variant proteins were used for the studies for this conclusion. G6PD is modified by multiple PTMs, the E coli-expressed protein may not fully represent mammalian cell-expressed G6PD.

The author stated that "However, the effect and significance of individual acetylation sites are still unknown or partially inferred from Lys-to-Glu mutational analysis". This statement is not accurate. For example, Shan et al., showed that G6PD K76 and K294 activates G6PD; Wang et al., 2014, determined that G6PD activity is negatively regulated by K403 via SIRT and KAT9, and K403 G6PD cannot form dimers. Etc.

The authors stated that "Our data underscore the complex roles of acetylation as a posttranslational modification that orchestrates the regulation of enzymatic activity, posttranslational modifications, and apoptotic signaling" G6PD PTM regulates PTMs in a feedback mechanism? A secondary PTM modification? No data supports this statement.

What is the rationale of the thermodynamic stability of acetylated G6PD protein? Substitution of lysine with another amino acid is an artificial model, which may be ideal for functional study, but the substituted amino acid is not natural for G6PD, the structural study's relevance to G6PD physiological function is not clear.

Reviewer #1 (Remarks to the Author)

Comment:

The manuscript by Wu et al, presents a very thorough study of lysine acetylation of glucose-6-phosphate dehydrogenase and its role on the regulation of enzymatic activity, and apoptotic signaling. Their conclusions are adequately supported by enzyme kinetics, immunochemistry, immunofluorescence, cellular, X-ray crystallography, and proteomics data which is technically sound. It was a great pleasure to read such methodologically precise and well written paper, I do not have any comments, and I am happy to recommend its publication.

Reply:

We thank the Reviewer for their positive feedback.

Reviewer #2 (Remarks to the Author)

In the article “Acetylation-dependent coupling between G6PD activity and apoptotic signaling”, Wu and co-workers investigate the molecular details of how acetylation at specific lysine residues modulates the activity of glucose-6-phosphate dehydrogenase (G6PD). Previous proteomics work has identified many acetylated lysines (AcKs) on G6PD under various condition. Most biochemical follow-up research has relied on site-directed mutagenesis to mimic acetylated (glutamine) or deacetylated (arginine) lysine. As Wu et. al. note, however, glutamine can sometimes be a poor mimetic for AcK. After prioritizing a handful of lysines based on multiple criteria, the authors use the amber codon method to incorporate AcK into G6PD at 100% stoichiometry onto specific sites. By performing both cell-based and in vitro experiments, they demonstrate opposing effects of AcK on two different sites in inhibiting (K403) and activating (K89) the enzyme. Furthermore, the authors show that the two contrasting acetyl marks participate in cross-talk with ubiquitination and phosphorylation. This work does a nice job of carefully characterizing complex PTM-mediated regulation of G6PD, which is quite complicated do to the PTM crosstalk. As this is an important enzyme for cancer biology and other research fields, I believe that the data will be of wide interest. My only primary concern is that the authors conclusions from acetyl immunoblots early in the paper assume that there is no crosstalk among acetyl sites, even though their model at the end of the paper suggests that G6PD is subject to extensive PTM crosstalk. A few relatively straight-forward mass spectrometry analyses would shore up this issue. Therefore, I recommend that the manuscript be reevaluated for publication in Nature Communications after the authors address a small number of my concerns and those of the other reviewers.

Primary Concerns

Comment:

1) Page 5 Line 22: How can the authors know for certain that all of the detected acetylation is indeed from the non-natural amino acid incorporation in each case? Is it not possible that

some of the signal for specific AcK variants in Figure 1C is from endogenous acetylation on other lysines than the one containing the non-natural AcK? The authors show later in the paper that PTM crosstalk exists between AcK and ubiquitination and phosphorylation— couldn't it also exist between AcK on different sites by acetylation of one residue recruiting/repelling an acetyltransferase or deacetylase? I suggest the authors conduct unbiased LC-MS/MS analysis of at least a subset of these acetylated proteins treated with and without KDACi to support this assumption that the signal detected by the western blots is strictly from the single AcK sites.

Reply:

We thank the Reviewer for highlighting the possibility of acetylation-dependent acetylation. Our manuscript includes activity measurements performed with site-specifically acetylated G6PD expressed in mammalian cells as well as in bacteria (where we do not expect to find acetylation-dependent acetylation). Based on the agreement between the measurements, we conclude that the observed differences in catalytic activity are due to a specific acetylation event. That said, we agree with the Reviewer that we cannot rule out acetylation-dependent acetylation events in mammalian cells, that have no effect on catalytic activity.

Following the Reviewer's comment, we performed LC-MS/MS analysis of relevant G6PD variants that were studied in our manuscript, including WT G6PD, following expression in cultured mammalian cells incubated in the presence or absence of KDACi. Our data show no secondary acetylation-dependent acetylation events. The new data is now summarized in Supplementary Table 1, and also mentioned in the manuscript (page 4):

Moreover, LC-MS/MS analyses of immunopurified G6PD expressed in the presence or absence of KDACi confirmed that the genetically encoded acetylated lysine residues are the most abundant acetylation sites, with no additional acetylation-dependent acetylation events (Supplementary Table 1).

Minor Concerns

Comment:

2) Page 2, Line 6: The authors may want to revise this sentence to read “Originally thought to be exclusively involved....acetylation is now additionally associated....”, to make it clear that the idea that acetylation regulates histones has not changed, but acetylation is now known to also have a broader role in the cell.

Reply:

We thank the Reviewer for their suggestion and revised the sentence accordingly (page 1):

Originally thought to be involved in transcription regulation via the acetylation of histone proteins, acetylation is now additionally associated with metabolism, cellular signaling, and other major cellular functions.

Comment:

3) Page 3, Line 15: Non-canonical amino acid incorporation is technically not dependent on there being evidence that the a PTM exists in vivo, but the rationale for developing hypotheses to test using previous data makes sense. Therefore, I suggest changing “known acetylation sites” to “any lysine, most commonly those already known to be acetylated from previous data.”

Reply:

We thank the Reviewer for highlighting this unclarity in our text. We corrected the text, and the revised version of the sentence now reads (page 3):

The charged tRNA_{ACUA} enables the cotranslational incorporation of AcK in response to an in-frame UAG (amber) stop codon mutation placed at essentially any position along the expressed protein of interest, most commonly, at lysine residue already known to be acetylated from previous data.

Comment:

4) Page, 4, Line 12: It was unclear to me how the pWT variants serve as appropriate controls, as “AcK was incorporated into a permissive site”. I’m sure that I am missing something, but the authors should clarify by adding an additional sentence explaining how this works. It seems important that the pWT does not have any AcK incorporated as I see no signal on the western blot in Figure 1C.

Reply:

The concept of using pWT as a control, which we introduced in this manuscript, is essential to our work as it is our reference point for the non-acetylated state. Therefore, we thank the Reviewer for bringing to our attention the lack of a proper explanation for the nature of our main control measurement. To better explain the idea of using pWT variants, we added the following explanation (page 4):

In pWT G6PD, Q83 or N414 are mutated to AcK. Therefore, these variants are expressed at the same level as other physiologically relevant acetylated variants and can serve as a non-acetylated control. In addition, the use of pWT as a control takes into account any unknown effects of amber suppression or expression of truncated proteins, on cell physiology.

Comment:

5) Page 15, Line 48: While the authors did not identify a peptide including Y401 and K403, was this only tried with trypsin? This experiment could be repeated using another protease that may produce a more easily detectable peptide spanning these residues.

Reply:

We agree with the Reviewer that a different protease might resolve this technical problem. Unfortunately, despite our attempts, we could not detect by mass spectrometry the peptide corresponding to Y401 and AcK403. Consequently, we made it clear that we cannot draw a conclusion about the phosphorylation state of Y401 in AcK403 G6PD.

Comment:

6) The authors are quite thorough in referencing appropriate work. Several missing papers that the authors could consider citing include a review that describes regulation of G6PD by PTMs (PMID: 36091812) and several primary articles on G6PD regulation by acetylation (PMID: 24769394; PMID: 27356773; PMID: 34310804).

Reply:

We thank the Reviewer for bringing these manuscripts to our attention. In addition to the work by Wang et al. (PMID: 24769394) that was included in our original submission (reference no. 34), the revised version of our manuscript now includes the other three papers:

PMID 36091812: Reference no. 17.

PMID 27356773: Reference no. 33.

PMID 34310804: Reference no. 37.

Comment:

7) Figure 7 legend: This figure is a nice visual summary of the study. However, the figure legend should be made to briefly describe all aspects of what the figure shows. This figure is very important for the reader to understand the take home from a lot of data.

Reply:

To explain the information summarized in Figure 7, we added the following explanation to the figure legend:

Suggested model for the effect of K89 and K403 acetylation on activity and cellular functions of G6PD. Left: Acetylation of K89 enhances the catalytic activity of G6PD and promotes its ubiquitylation on residues K95/K97. Sirt1 and Sirt2 can deacetylate AcK89. **Right:** K403 is acetylated by KAT9 (Wang et al.³⁴) and CBP (this work) and deacetylated by Sirt2 (Wang et al.³⁴ and this work) and Sirt1 (this work). Acetylation of K403 inhibits G6PD activity and promotes the interaction with p53, leading to the stabilization of p53 and induction of proapoptotic signaling. K403 acetylation also promotes Fyn-dependent phosphorylation of Y503, which can be partially inhibited by the interaction between K403-acetylated G6PD and p53. Y503 phosphorylation has no effect on K403 deacetylation by Sirt1 and Sirt2.

Reviewer #3 (Remarks to the Author)

The manuscript by Wu et al., entitled "Acetylation-dependent coupling between G6PD activity and apoptotic signaling" reports the link between lysine acetylation-dependent G6PD activity and tumor cell apoptotic signaling. The authors first sought to determine the effect of lysine acetylation on G6PD activity using deacetylase inhibitors and determined that sirtuin-mediated lysine acetylation modulates G6PD activity. The authors then investigated the role of each of the 9 known acetylated lysines in G6PD using the point/site-mutated G6PD variants in G6PD knocked-down tumor cell lines. K403 was found to reduce, whereas K89

was found to increase G6PD enzymatic activity, and the balance of K89, K171, K386, K403, K408, and K432 modules G6PD activity. Using E. coli expressed and purified G6PD variants, the author concluded that G6PD can be activated by a single lysine acetylation but is de-activated by multiple lysine acetylation. The author then studied the stability of the recombinant G6PD mutants and aggregation. The authors focused on K89 and K403, particularly K403 for its regulation by KATs and KDACs and its effects on G6PD conformations and phosphorylation, and determined that K403Ac deforms the dimer interface and renders G6PD monomeric. Further studies determined that K403Ac promotes p53 interaction and G6PD to stabilize p53. Finally, the authors observed that G6PD K403Ac promotes Bax accumulation through a p53-dependent mechanism, which leads to the conclusion that G6PD K403 acetylation promotes the induction of apoptotic signaling.

Overall, the authors performed extensive biochemical and structural studies to determine lysine acetylation and G6PD function. PTM, including lysine acetylation, plays an important role in G6PD function, the finding that K403 of G6PD may be important for G6PD function in tumor cell apoptosis is interesting.

Major concerns

Comment:

The major concern is the lack of new findings. All the studied lysine acetylation sites in G6PD are known in the literature. K403 acetylation in G6PD has been extensively studied as a negative regulator of G6PD activity. P53 also has been shown to bind to G6PD to prevent G6PD dimerization. The authors state that they found that K403 acetylation promotes the interaction between p53 and G6PD and stabilizes p53 in cells. However, what is new here in terms of G6PD dimer formation and function is not clear.

Reply:

Our study is based on putative lysine acetylation sites identified by different MS proteomic studies and not functional studies (Supplementary Figure 1b). The positions are known, but except for K403, the functional consequences of acetylation at all the other positions were not studied at all, or were studied using mutated G6PD (i.e., K171). As mentioned and acknowledged in our manuscript (e.g., reference no. 33 and the new legend to Figure 7), K403 acetylation is known to inactivate G6PD and render the protein monomeric. While confirming these observations, our manuscript includes new findings: (1) The crystal structure of K403-acetylated G6PD, which provides the molecular mechanism of acetylation-dependent inhibition and disruption of the dimeric structure; (2) We identify Sirt1 and CBP as deacetylase and acetyltransferase of AcK403/K403; (3) We identify Y503 as a substrate of Fyn kinase and show that K403 acetylation promotes Y503 phosphorylation; (4) We show that K403 acetylation stabilizes the (previously identified) interaction between G6PD and p53. This observation begs the question: is the interaction with p53 renders G6PD monomeric and inactive (as mentioned by the Reviewer), or maybe the population of G6PD that favorably interacts with p53 is monomeric and inactive due to K403 acetylation?; (5) We show that K403 acetylation promotes p53-dependent pro-apoptotic signaling; (6) We found that K89 acetylation enhances the catalytic activity of G6PD, relative to the substrate

NADP⁺. (7) We found that K89 acetylation promotes the ubiquitylation of K95/K97 and determined the crystal structure of K89-acetylated G6PD, which suggests a possible explanation for the observed acetylation-dependent ubiquitylation.

While the acetylation of G6PD has been studied in the past, we strongly believe that our work provides new information regarding the functional roles of G6PD acetylation, as well as mechanistic aspects.

Comment:

A major conclusion of this manuscript is the G6PD-p53-Bax-apoptotic signaling. The authors observed an increase in Bax in AcK403 G6PD-expressed cells, but not in pWT G6PD- cells. In parallel, HCT116 p53 WT cells have higher p53 than HCT116 p53 KO cells. These observations are correlative and no data to support the G6PD-p53-Bax-apoptosis pathway. Recent literature shows that there are compensatory pathways (ME1 and DH1) for G6PD and G6PD is not essential for certain tumor cell tumor cell proliferation and survival. A definitive experiment is to knock out G6PD to see if it is essential for tumor cell apoptosis here. In addition, G6PD/NADPH regulates HDAC3, a known apoptosis regulator. These may all affect the correlation observed in this study.

Reply:

The Reviewer raises a concern that we show a correlation between AcK403 acetylation and pro-apoptotic signaling. In their comment, the Reviewer suggests that causation should be proved by knocking-out G6PD to show that it is essential for tumor cell apoptosis. In addition, the Reviewer raises the possibility that the observed pro-apoptotic signaling is due to differences in NADPH levels that, in turn, regulate the apoptotic regulator HDAC3 (and not due to the reported G6PD-p53-Bax-apoptosis pathway). We appreciate the Reviewer's opinion and relate to these two concerns separately.

G6PD knockout:

As the Reviewer mentioned, we already demonstrated the role of p53 in the G6PD-p53-Bax-apoptosis pathway using a p53-knockout cell line. It is now suggested to demonstrate that G6PD is also essential for the observed induction of apoptosis by knocking out G6PD. We agree with the Reviewer that eliminating a given component is a logical way to prove that the component is essential for the observed phenotype. However:

1. Using G6PD that is site-specifically acetylated at different positions, we demonstrate that the essential pro-apoptotic event is K403 acetylation, not G6PD per se. Hence, the component that should be eliminated (or knocked out) is K403 acetylation, not G6PD. To check if K403 acetylation is essential for the observed pro-apoptotic signaling, we used the K403R mutant of G6PD that cannot be acetylated at this position, and the K403Q mutant that may mimic the acetylated protein. Compared to AcK403 G6PD, WT G6PD promoted a much weaker expression of Bax in both HEK293T and WT HCT116 cells. In both cell lines no increase in Bax or cleaved caspase 3 was observed following the expression of the K403R and even the K403Q mutants of G6PD. Thus, these data show

that K403 acetylation (and not G6PD level or activity) is essential for the observed pro-apoptotic signaling.

The new results are now presented in Supplementary Fig. 11c, and we refer to the new data on page 11:

These effects were not observed in HEK293T or WT HCT116 cells expressing the acetylation-resistant K403R G6PD mutant nor the acetyl-mimic K403Q mutant (Supplementary Fig. 11c), and cannot be attributed to differences in cellular NADPH levels (Supplementary Fig. 11d).

2. It has been shown that although G6PD knockout HCT116 cells can develop into subcutaneous xenografts, they have impaired growth and oxidative defense (PMID: 31058257, 32661137). Other studies have reported that inhibition of G6PD leads to decreased proliferation and renders the cells more prone to oxidative stress-induced apoptosis (PMID: 30066842, 11481225, 20032314). Therefore, G6PD knockout should not be used to determine if acetylation of G6PD is involved in pro-apoptotic signaling cascades since it is expected to negatively affect cell viability.
3. As the Reviewer mentioned, there are other, G6PD-independent, apoptotic signaling cascades in cells. Hence, G6PD is not 'essential for tumor cell apoptosis'. Therefore, we do not expect that knocking out G6PD will render the cells non-apoptotic. Moreover, knocking out G6PD will prevent us from checking the effect of G6PD acetylation, which we claim is the essential event in the suggested pathway.

NADPH and HDAC3:

We acknowledge the potential indirect effect of NADPH on apoptotic signaling via the regulation of histone acetylation by HDAC3 (PMID: 33462516). However, as the Reviewer mentioned, alternative pathways in the cell [i.e., malic enzyme 1 (ME1), and isocitrate dehydrogenase 1 (DH1)] can compensate for the lack of G6PD activity and maintain similar levels of NADPH, (Chen et al., Nat. Metab., 2019). Therefore, one should not expect significant differences in NADPH levels between cells expressing AcK403 or pWT G6PD, and, consequently, NADPH-dependent modulation of HDAC3 activity.

To verify that the observed K403 acetylation-dependent pro-apoptotic signaling is not caused by G6PD-dependent differences in NADPH concentration, we measured the cellular levels of NADPH. In line with previously published data (PMID: 26976705), the expression of exogenous active G6PD (pWT, AcK403, and AcK89) resulted in higher levels of NADPH. In contrast, the expression of inactive AcK171 G6PD had no effect on cellular NADPH levels. Importantly, we found no significant difference in NADPH levels between cells expressing pWT, AcK403, or AcK89 G6PD. We, therefore, conclude that the observed K403 acetylation-dependent pro-apoptotic signaling is independent of cellular NADPH levels. The new data is presented in Supplementary Fig. 11d, and described on page 11:

These effects were not observed in HEK293T or WT HCT116 cells expressing the acetylation-resistant K403R G6PD mutant nor the acetyl-mimic K403Q mutant

(Supplementary Fig. 11c), and cannot be attributed to differences in cellular NADPH levels (Supplementary Fig. 11d).

To further support our conclusion that the observed K403 acetylation-dependent pro-apoptotic signaling is not related to NADPH-dependent regulation of histone acetylation by HDAC3, we measured the levels of H3K9 and H3K18 acetylation (known substrates of HDAC3). As presented below, we found no difference in histone acetylation levels between cells expressing pWT, AcK171, or AcK403 G6PD, incubated with or without nicotinamide. These data support our conclusion that NADPH-dependent HDAC3 activity is not involved in the observed G6PD AcK403-p53-Bax apoptotic signaling. That said, considering the above-mentioned similar NADPH levels, we believe this measurement is not essential to establishing our conclusion. Therefore, we prefer not to include it in the final version of our manuscript.

Representative Western blot showing the levels of endogenous H3K9 and H3K18 acetylation, compared to total level of H3. Data suggest that levels of H3 acetylation is independent of the activity of exogenously expressed G6PD, nor its acetylation status or presence of nicotinamide (NAM).

Comment:

The author stated that: “Thus, using site-specifically acetylated G6PD, we show for the first time that G6PD can be activated and not only deactivated by the acetylation of a single lysine residue”. The term “for the first time” should be used with caution. There are extensive literatures reporting lysine acetylation and G6PD activity. Only a small part of related references is cited in this manuscript. In addition, E coli-expressed G6PD variant proteins were used for the studies for this conclusion. G6PD is modified by multiple PTMs, the E coli-expressed protein may not fully represent mammalian cell-expressed G6PD.

Reply:

We agree with the Reviewer that the term ‘for the first time’ should be used cautiously. Therefore, we removed it from this sentence (page 5):

Thus, using site-specifically acetylated G6PD, we show that G6PD can be activated and not only deactivated by the acetylation of a single lysine residue.

While it was known that K403 acetylation inhibits G6PD (as mentioned in the introduction), we provide evidence for the activation of G6PD by acetylation, that was not known before.

Importantly, our data is based on measurements performed with acetylated G6PD expressed in two different mammalian cell lines (e.g., Figure 1d), as well as with acetylated G6PD expressed in bacteria (e.g., Figure 2b). In both cases, K89 acetylated G6PD was more active than non-acetylated G6PD.

Comment:

The author stated that “However, the effect and significance of individual acetylation sites are still unknown or partially inferred from Lys-to-Glu mutational analysis”. This statement is not accurate. For example, Shan et al., showed that G6PD K76 and K294 activates G6PD; Wang et al., 2014, determined that G6PD activity is negatively regulated by K403 via SIRT and KAT9, and K403 G6PD cannot form dimers. Etc.

Reply:

We thank the Reviewer for their comment. As mentioned in our manuscript, proteomic studies identified several putative lysine acetylation sites on G6PD. The Reviewer is correct about the work by Wang et al., in which some of the experiments were performed with K403-acetylated G6PD. Therefore, we modified the text to reflect that (page 11):

However, the effect and significance of individual acetylation sites are mostly unknown or partially inferred from Lys-to-Gln mutational analyses.

It is important to mention that most of the data presented by Wang et al. were obtained using Lys-to-Gln mutants of G6PD, and the use of acetylated G6PD led us to different conclusions; for example, the effect of K89 acetylation, or the deacetylation of AcK403 by Sirt1.

The Reviewer cites the work by Shan et al. as an example of a study in which G6PD acetylation was studied using acetylated variants. However, the enzyme studied in this paper is 6PDG and not G6PD.

Comment:

The authors stated that “Our data underscore the complex roles of acetylation as a posttranslational modification that orchestrates the regulation of enzymatic activity, posttranslational modifications, and apoptotic signaling” G6PD PTM regulates PTMs in a feedback mechanism? A secondary PTM modification? No data supports this statement.

Reply:

In Figure 3 we provide evidence for K89 acetylation-dependent ubiquitylation of G6PD. In addition, in Figure 6 and Supplementary Figure 8, we show that K403 acetylation promotes the phosphorylation of Y503. While we do not suggest that these PTM-dependent PTMs participate in a feedback mechanism, we show that site-specific acetylation promotes the PTM of other residues.

That said, following the Reviewer’s comment, we edited the above-mentioned sentence in the Abstract, so it will be clear that we provide an example for the complex roles of acetylation (Abstract, page 2):

Our data provide an example of the complex roles of acetylation as a posttranslational modification that orchestrates the regulation of enzymatic activity, posttranslational modifications, and apoptotic signaling.

Comment:

What is the rationale of the thermodynamic stability of acetylated G6PD protein? Substitution of lysine with another amino acid is an artificial model, which may be ideal for functional study, but the substituted amino acid is not natural for G6PD, the structural study's relevance to G6PD physiological function is not clear.

Reply:

Many mutations that lead to severe cases of G6PD deficiency trigger structural defects at the dimer interface and the active site. These structural distortions not only inhibit G6PD activity but also destabilize the enzyme. Hence, the effect of acetylation-dependent electrostatic and steric changes on the structure, thermodynamic stability, and activity of G6PD, are significant. Importantly, a major advantage of our study is that we study site-specifically acetylated G6PD variants and not mutants of G6PD (as suggested by the Reviewer).

Acetylation is a natural modification of G6PD, and our studies provide the most accurate and physiologically relevant data related to the effect of acetylation on the structure, thermodynamic stability, and function of G6PD.

REVIEWERS' COMMENTS

Reviewer #2 (Remarks to the Author):

The authors have satisfactorily addressed my concerns and I believe their improved manuscript will be of wide interest to the metabolism and cell signaling research fields. I have a couple final suggestions for the authors to consider, but leave it to them to decide if addressing these would add to the final version of their manuscript. I recommend the manuscript be accepted for publication in Nature Communications, pending remaining concerns from the other reviewers.

Final Suggestions:

1) It may be helpful to include some discussion of key residues of acetylation, phosphorylation, and ubiquitination in the abstract. As there are a lot of PTM sites discussed in the results section, including those that are found to have the most significant regulatory roles could help the abstract better convey the take-home message (similar to the nice changes to the legend of Figure 7). Obviously, this would need to be very concise.

2) It would also be helpful to include Y503 in Supplementary Figure 1A, perhaps using another color to indicate that this is a site of phosphorylation. Similarly, K95 and K97 could be highlighted in some way to indicate that these are the residues regulated by ubiquitination--maybe using red and underlining to highlight these lysines as sites of acetylation and ubiquitination, respectively. Including all sites of PTM assessed in the manuscript would add significantly to this figure and make it serve as a more detailed complement to the nicely succinct Figure 7--maybe even including something like "see Supplementary Figure 1A for all sites of PTM investigated" in the Figure 7 legend.

Reviewer #3 (Remarks to the Author):

The authors have made an good effort to address this reviewer's comments. Although the new experiments are not performed, the authors made scientifically sounded rebuttal and clarifications. Most of my concerns have been addressed through clarifications in the revised manuscript. This reviewer has no more concerns.

Reviewer #2 (Remarks to the Author)

The authors have satisfactorily addressed my concerns and I believe their improved manuscript will be of wide interest to the metabolism and cell signaling research fields. I have a couple final suggestions for the authors to consider, but leave it to them to decide if addressing these would add to the final version of their manuscript. I recommend the manuscript be accepted for publication in Nature Communications, pending remaining concerns from the other reviewers.

Comment:

1) It may be helpful to include some discussion of key residues of acetylation, phosphorylation, and ubiquitination in the abstract. As there are a lot of PTM sites discussed in the results section, including those that are found to have the most significant regulatory roles could help the abstract better convey the take-home message (similar to the nice changes to the legend of Figure 7). Obviously, this would need to be very concise.

Reply:

We thank the Reviewer for their suggestion. The positions of key posttranslationally modified residues are now mentioned in the abstract. A brief discussion, as suggested by the Reviewer, would be more informative, but we had to compromise to comply with the limitation of 150 words.

Comment:

2) It would also be helpful to include Y503 in Supplementary Figure 1A, perhaps using another color to indicate that this is a site of phosphorylation. Similarly, K95 and K97 could be highlighted in some way to indicate that these are the residues regulated by ubiquitination--maybe using red and underlining to highlight these lysines as sites of acetylation and ubiquitination, respectively. Including all sites of PTM assessed in the manuscript would add significantly to this figure and make it serve as a more detailed complement to the nicely succinct Figure 7--maybe even including something like "see Supplementary Figure 1A for all sites of PTM investigated" in the Figure 7 legend.

Reply:

We thank the Reviewer for their suggestion and modified Supplementary Figure 1A accordingly. All the posttranslationally modified residues discussed in the manuscript (including tyrosine phosphorylation and lysine ubiquitylation sites) are included in the new version of the figure and distinguished from one another by a color code.

Reviewer #3 (Remarks to the Author):

The authors have made an good effort to address this reviewer's comments. Although the new experiments are not performed, the authors made scientifically sounded rebuttal and

clarifications. Most of my concerns have been addressed through clarifications in the revised manuscript. This reviewer has no more concerns.